# Seismic Response of RC Beam-Column Joints Strengthened with FRP ROPES, Using 3D Finite Element: Verification with Real Scale Tests

**Emmanouil Golias** [1], **Paul Touratzidis** [1] **and Chris G. Karayannis** [2,*]

1   Laboratory of Reinforced Concrete and Seismic Design of Structures, Civil Engineering Department, Democritus University of Thrace, 67100 Xanthi, Greece; egkolias@civil.duth.gr (E.G.); pavlostur@gmail.com (P.T.)
2   Laboratory of Reinforced Concrete Structures and Masonry Structures, Civil Engineering Department, Aristotle University of Thessaloniki, 54636 Thessaloniki, Greece
*   Correspondence: karayannis@civil.auth.gr

**Abstract:** A 3D-finite element analysis within the numerical program ABAQUS is adopted in order to simulate the seismic behavior of reinforced concrete beam-column joints and beam-column joints strengthened with CFRP ropes. The suitability of the adopted approach is investigated herein. For this purpose, experimental and numerical cyclic tests were performed. The experiments include four reinforced concrete (RC) joints with the same ratio of shear closed-stirrup reinforcement and two different volumetric ratios of longitudinal steel reinforcing bars. Two joints were tested as-built, and the other two were strengthened with CFRP ropes. The ropes were applied as Near Surface Mounted (NSM) reinforcement, forming an X-shape around the joint body and further as flexural reinforcement at the top and bottom of the beam. The purpose of the externally mounted CFRP ropes is to allow the development of higher values of concrete principal stresses inside the joint core, compared with the specimens without ropes, and also to reduce the developing shear deformation in the joint. From the results, it is concluded that X-shaped ropes reduced the shear deformation in the joint body remarkably, especially in high drifts. Further, as a result of the comparisons between the yielded outcome from the attempted nonlinear analysis and the observed response from the tests, it is deduced that the adopted method sufficiently describes the whole behavior of the RC beam-column connections. In particular, comparisons between experimental and numerical results of principal stresses developing in the joint body of all examined specimens, along with similar comparisons of force displacement envelopes and shear deformations of the joint body, confirmed the adequacy of the applied finite element approach for the investigation of the use of CFRP-ropes as an efficient and easy-to-apply strengthening technique. The findings also reveal that the connections that have been strengthened with the FRP ropes demonstrated improved performance, and the crack system preserved its load capacity during the reversal loading tests.

**Keywords:** reinforced concrete joints; FRP ropes; 3D finite elements; NSM strengthening of RC elements; cyclic loading

## 1. Introduction

Beam-column connections in RC structures have systematically been proven to be critical elements for the seismic response of buildings. Therefore, for RC beam-column joints that have been designed and constructed according to previous regulations, it is a common practice to be strengthened. Therefore, applied methods and practices for the strengthening of beam-column joints are very important. Recently, the application of FRP ropes as an easy-to-apply method for the strengthening of RC elements has repeatedly been reported in science literature [1–8]. For the study of this innovative method, obviously, the most appropriate research direction is the experimental investigation [1,3,5]. However, inherent problems concerning the experimental

studies, mainly the increased expenses, and on the other hand, the significant advancement of the numerical and numerical procedures have proved the application of 3D finite elements as an effective alternative approach very convenient for the investigation of the seismic response of the beam-column connections [9–19].

Concrete as a material has a nonlinearity that can be approached using various models like nonlinear elasticity, elastoplastic model, or even special damage models. Although nonlinear elasticity and elastoplastic models successfully simulate the loading branch of the concrete stress–strain relationship, they cannot satisfactorily predict the material behavior under cyclic loading since, in this case, accumulation of plastic strain and stiffness degradation are simultaneously observed.

In this direction, the Concrete Damaged Plasticity (CDP), as implemented in the well-established computer program ABAQUS, is a very suitable model since it can take into account both material damage and plastic strain [20]. This model has been successfully applied to predict the behavior of concrete elements under various loading situations and geometrical conditions.

The stress–strain relationship for steel was modeled using an elastic-perfectly plastic model, while the behavior of carbon fiber reinforced polymer (CFRP) ropes, utilized for reinforcement, was represented with a linear elastic model due to their limited capacity for elastic tension. This comprehensive modeling approach allowed for the accurate simulation of the unique response of reinforced concrete elements under cyclic loading conditions.

Further, there are papers in the literature where finite elements, and in particular the concrete damage plastic model, have been applied to study reinforced concrete elements or structures under static or dynamic loading. Among them, there are works where comparisons between tested data and numerical results indicated remarkable consistency, proving that the Concrete Damaged Plasticity model can be considered a robust model for the study of concrete elements [20–22].

Numerical simulations offer a cost-effective means to study complex structural behavior. ABAQUS, a widely-used finite element analysis software, provides robust capabilities for modeling concrete and steel behavior under various loading conditions; recent literature indicated the efficiency of finite elements and numerical analysis with simulation via ABACUS (Shakor et al. [23]).

It is worth noting that ABAQUS offers distinct advantages over other finite element software due to its robust capabilities for modeling nonlinear material behavior, complex geometries, and dynamic loading conditions.

Therefore, the Concrete Damaged Plasticity model has been adopted for the simulation of concrete. The stress–strain relationship adopted for steel is an elastic-perfectly plastic model. Finally, the stress–strain relationship applied for the Carbon Fiber Reinforced Polymer (CFRP) ropes include only the linear elastic part since it can only sustain elastic tension [22–27]. From the results, it can be deduced that the numerical force displacement outcome is very close to the experimental one, and therefore, all adopted damage laws proved successful in simulating the unique response of reinforced concrete under cyclic loading.

The geometrical characteristics of the specimens were meticulously designed to resemble common structures encountered in real-world applications. This design approach aimed to ensure the relevance and applicability of the experimental findings to a broader spectrum of structural configurations typically found in practice. As mentioned before, the dimensions, the reinforcements, and the overall design of these specimens comply with the corresponding ones of external beam-column connections of a multi-story reinforced concrete frame structure [28,29]. Therefore, it is concluded that limitations can be considered only regarding the dimensions of the tested elements. Based on these considerations and given the limited existing research in this area, the examined strengthening technique, with the use of CFRP ropes, has to be further investigated experimentally and numerically based on tested specimens with substantially larger dimensions. However, while the specimens were designed to emulate common structural configurations, it is essential to acknowledge

that the study's findings may have limitations in fully capturing the complexity and variability inherent in real-world structures. Factors such as variations in material properties, construction practices, and environmental conditions could influence the performance of actual structures differently than observed in controlled laboratory settings. Therefore, while the specimens offer valuable insights, their representativeness for a wider range of real-world applications should be considered within the context of specific structural designs and environmental conditions.

## 2. Materials and Methods

### 2.1. Characteristics of the Specimens

The geometry and reinforcements of the specimens have been chosen to be similar to the geometrical characteristics and reinforcements of columns and beams (Figure 1) of common structures. The total length of the column part of the specimens is equal to 3.0 m, and its cross-section is 350/250 mm, whereas the length of the beam is 1.875 m, and its cross-section dimensions are 350/250 mm. Reinforcements are presented in Figures 1 and 2, whereas the amounts of reinforcements are given in Table 1. The main purpose of the experimental project is to investigate the effectiveness of the application of near-surface mounted (NSM) CFRP ropes diagonally placed on the two sides of the joint body as a strengthening technique [1,24,25]. Therefore, four specimens are tested: Specimen JA0 and JA1 as pilot specimens and specimens JA0Fxb and JA0F2x2b (Figure 2) strengthened with diagonally placed CFRP ropes. Locations of the CFRP ropes applied as NSM strengthening reinforcement in specimens JA0Fxb and JA0F2x2b are presented in Figures 2a and 2b, respectively.

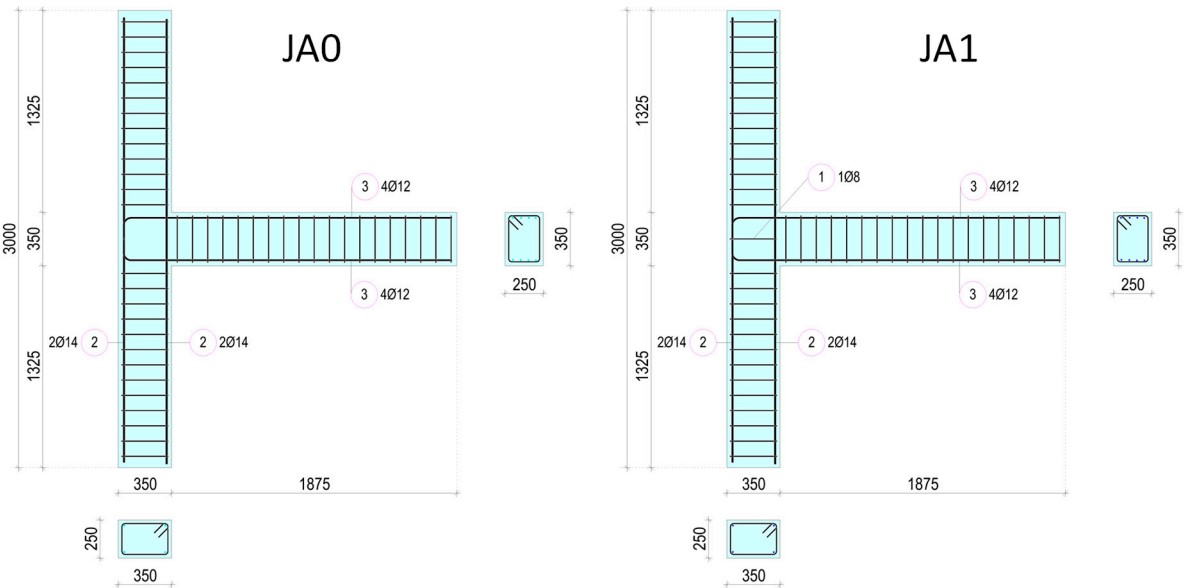

**Figure 1.** Geometrical characteristics of specimens JA0 and JA1 (dimensions in mm) and steel reinforcements (see also Table 1).

**Table 1.** Reinforcements of beam-column specimens.

| Reinforcements | JA0 | JA1 | JA0Fxb | JA0F2x2b |
|---|---|---|---|---|
| ① | - | 1Ø8 | - | - |
| ② | 2Ø14 | 2Ø14 | 2Ø14 | 2Ø14 |
| ③ | 4Ø12 | 4Ø12 | 4Ø12 | 4Ø12 |
| FRP ropes of joint | - | - | X-type Single rope | X-type Double rope |
| FRP ropes of beam | - | - | Single rope | Double rope |

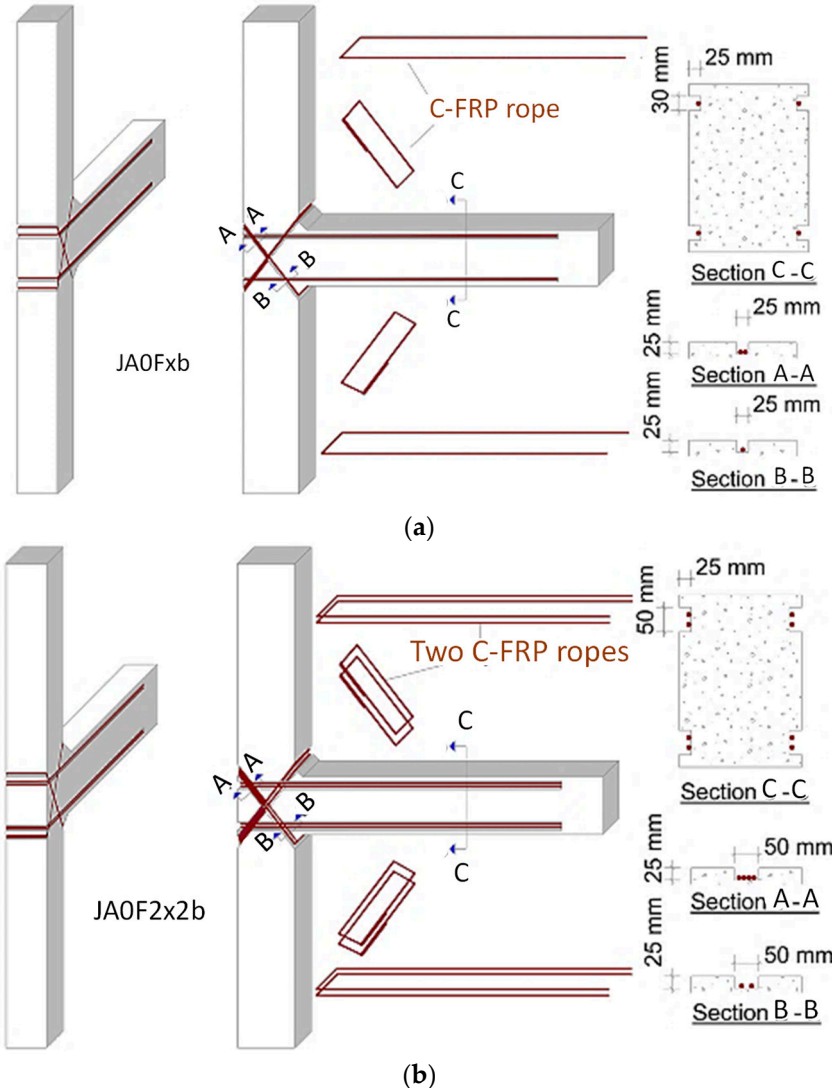

**Figure 2.** Location of the CFRP ropes applied as NSM strengthening reinforcement in specimens JA0Fxb and JA0F2x2b. (**a**) CFRP ropes of Specimen JA0Fxb. (**b**) CFRP ropes of Specimen JA0F2x2b.

It is emphasized that the experimental project conducted for this study includes five full-scale exterior beam-column connections constructed and tested under increasing cyclic loading. The dimensions, the reinforcements, and the overall design of these specimens comply with the corresponding ones of external beam-column connections of the upper floors of a multi-story reinforced concrete frame structure. Therefore, it is concluded that limitations can be considered only regarding the dimensions of the tested elements; thereupon, it is a fact that the extracted conclusions based on the test results may not be entirely applied to joints of the lower floors of tall structures if the dimension is substantially larger than the considered ones. Based on these considerations and given the limited existing research in this area, the examined strengthening technique, with the use of CFRP ropes, has to be further investigated experimentally and numerically based on tested specimens with substantially larger dimensions.

While the manuscript presents the CFRP rope technique as an effective method for strengthening reinforced concrete beam-column joints, a more comprehensive comparison with alternative strengthening approaches alleges significant advantages over the other techniques. Various contemporary methods, such as reinforced concrete jackets, external steel plates, CFRP sheets, and steel jacketing, offer alternative solutions for enhancing the seismic performance of concrete structures. Nevertheless, in terms of cost, the CFRP

rope technique may offer significant advantages over traditional methods like reinforced concrete jacketing and steel jacketing, which apparently require significantly higher labor and material costs for installation. Moreover, the implementation complexity of the CFRP rope technique, involving the application of ropes in superficial notches, appears to be much simpler and less invasive compared to techniques like reinforced concrete or shotgun concrete jacketing, which involve extensive retrofitting and structural modifications. However, the effectiveness of the CFRP rope technique in improving the seismic response of beam-column joints should be further investigated against these alternatives. While the manuscript highlights the benefits of the CFRP rope technique, the presented numerical and tested results of its effectiveness in terms of load-carrying capacity, ductility enhancement, and crack mitigation provide valuable insights for structural engineers and practitioners.

Carbon fiber-reinforced polymer ropes are composite materials composed of high-strength carbon fibers embedded in a polymer matrix, typically epoxy resin. The physical properties of CFRP ropes are characterized by their high tensile strength, stiffness, and low weight, making them ideal for structural reinforcement applications. The carbon fibers provide excellent mechanical properties, with tensile strengths ranging from 2000 MPa to 7000 MPa, and modulus of elasticity ranging from 200 GPa to 800 GPa, depending on the manufacturing process and fiber orientation; the mechanical characteristics of the CFRP ropes used in this study as given by the manufacturer are: tensile strength 4000 MPa, tensional modulus of elasticity 240 GPa and cross-section of Carbon fibers $A_f$ = 28 mm$^2$. CFRP ropes also exhibit excellent corrosion resistance and durability, making them suitable for application in harsh environmental conditions. Additionally, CFRP ropes can be easily fabricated into various shapes and configurations, allowing for flexible application in strengthening structural elements such as beam-column joints in reinforced concrete structures. The chemical properties of CFRP ropes are primarily determined using the epoxy resin matrix, which provides adhesion between the carbon fibers and protects them from environmental degradation. Epoxy resins offer high chemical resistance, low shrinkage during curing, and excellent bonding properties, ensuring the long-term performance and durability of CFRP strengthening systems.

### 2.2. Concrete Damage Plasticity

The Concrete Damage Plasticity model, as implemented in ABAQUS, is suitable for simulating almost all kinds of concrete elements like columns, beams, and trusses. It is based on the concept of isotropic elasticity in combination with plasticity in compression and tension. The same finite element can also be applied to simulate the steel bars of the reinforcement. The Concrete Damaged Plasticity model is mainly presented in work by Lubliner et al., and further, it has been extended by Lee and Fenves [30,31]. It takes into account plasticity, softening, and damage features, which are also applicable to cyclic loading. A Drucker-Prager hyperbolic function is incorporated in the yield function (Figure 3) [20,32].

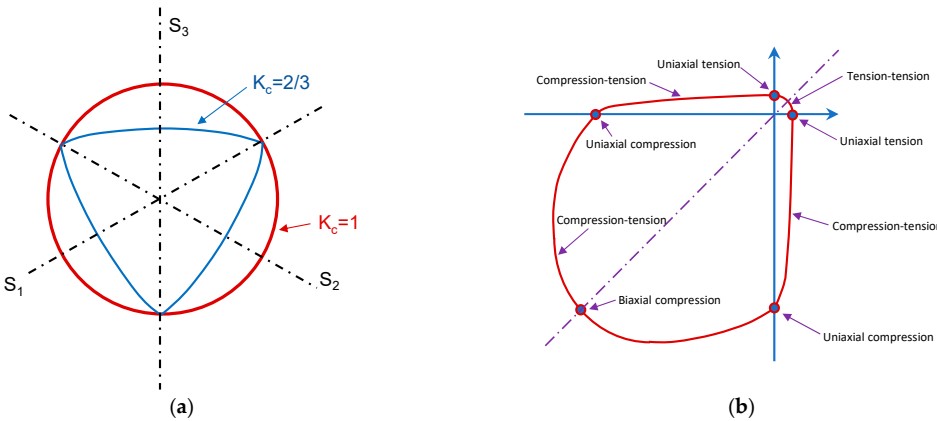

(a)                                                                 (b)

**Figure 3.** Yield surfaces (**a**) in the deviatoric plane correspond to different values of $K_c$ and (**b**) in plane stress.

### 2.2.1. Yield Function

As mentioned before, the adopted model uses the yield function of Lubliner et al. [31], with the modifications proposed by Lee and Fenves to account for different evolutions of strength under tension and compression [30]. The yield function is controlled by the hardening variables $\bar{\varepsilon}_t^{pl}$ and $\bar{\varepsilon}_c^{pl}$ and the yield function takes the following form:

$$F = \frac{1}{1-a}\left(\bar{q} - 3a\bar{p} + \beta\left(\bar{\varepsilon}^{pl}\right) <\hat{\bar{\sigma}}_{max}> -\gamma<-\hat{\bar{\sigma}}_{max}>\right) - \bar{\sigma}_c\left(\bar{\varepsilon}_c^{pl}\right) = 0 \tag{1}$$

where

$$\alpha = \frac{\left(\frac{\sigma_{bo}}{\sigma_{co}}\right) - 1}{2\left(\frac{\sigma_{bo}}{\sigma_{co}}\right) - 1} \quad 0 \le a \le 0.5 \tag{2}$$

$$\beta = \frac{\bar{\sigma}_c\left(\varepsilon_c^{-pl}\right)}{\bar{\sigma}_t\left(\varepsilon_t^{-pl}\right)}(1-\alpha) - (1+\alpha) \tag{3}$$

$$\gamma = \frac{3(1-K_c)}{2K_c - 1} \tag{4}$$

$\hat{\bar{\sigma}}_{max}$ is the maximum principal effective stress;

$\left(\frac{\sigma_{bo}}{\sigma_{co}}\right)$ is the ratio of initial biaxial compressive yield stress to initial uniaxial compressive yield stress,

$K_c$ is the ratio of the second stress invariant on the tensile meridian, $q_{(TM)}$, to that on the compressive meridian, $q_{(TM)}$, at initial yield for any given value of the pressure invariant $p$ such that the maximum principal stress is negative, $\hat{\bar{\sigma}}_{max} < 0$,

$\bar{\sigma}_t\left(\bar{\varepsilon}_t^{pl}\right)$ is the effective tensile cohesion stress and $\bar{\sigma}_c\left(\bar{\varepsilon}_c^{pl}\right)$ is the effective compressive cohesion stress.

### 2.2.2. Uniaxial Tension and Compression Stress Behavior

The uniaxial tension and the uniaxial compression stress–strain relationships are characterized by damaged plasticity, as shown in Figure 4.

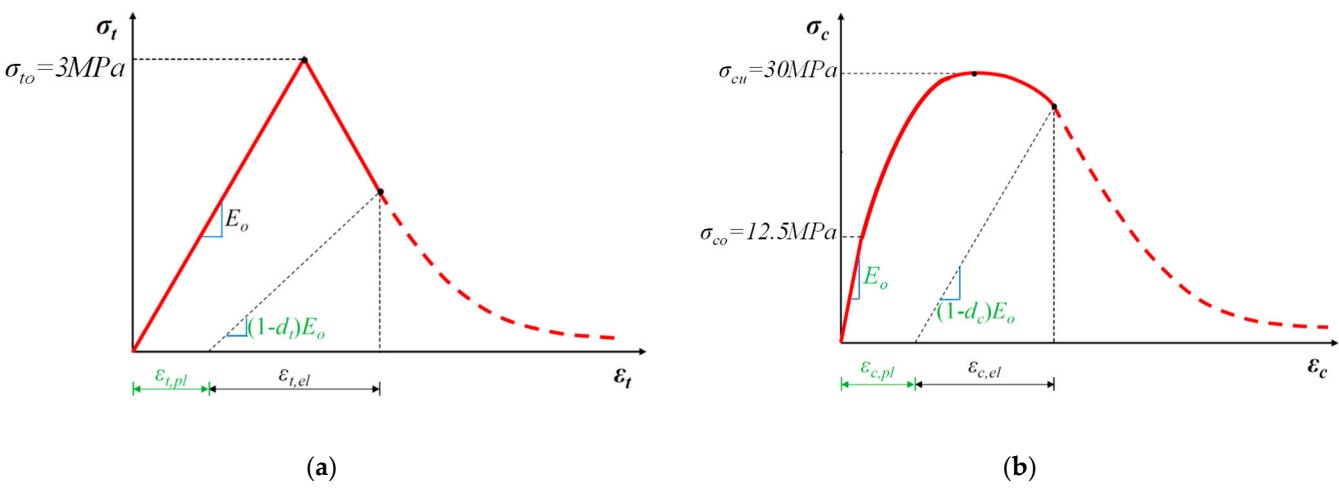

(**a**)          (**b**)

**Figure 4.** Response of concrete to uniaxial loading, (**a**) in tension and (**b**) in compression.

In the case of uniaxial tension, a linear elastic branch is adopted until cracking stress, $\sigma_{to}$, (Figure 4a), which corresponds to the onset of micro-cracking in the concrete material. In the case of uniaxial compression, a linear elastic branch is adopted until initial yield stress, $\sigma_{co}$, (Figure 4b). The inelastic part of the response includes stress hardening followed by strain softening beyond the maximum stress, $\sigma_{cu}$. Thus, the adopted stress–strain

relationship includes the main features of the response of concrete [20,32,33]. The uniaxial stress–strain curves are converted to stress versus plastic-strain curves.

Thus,

$$\sigma_t = \sigma_t\left(\bar{\varepsilon}_t^{pl}, \dot{\bar{\varepsilon}}_t^{pl}, f_i\right) \tag{5}$$

$$\sigma_c = \sigma_c\left(\bar{\varepsilon}_c^{pl}, \dot{\bar{\varepsilon}}_c^{pl}, f_i\right) \tag{6}$$

where subscripts t and c refer to tension and compression, respectively, $\bar{\varepsilon}_t^{pl}$ and $\bar{\varepsilon}_c^{pl}$ are the equivalent plastic strains, $\dot{\bar{\varepsilon}}_t^{pl}$ and $\dot{\bar{\varepsilon}}_c^{pl}$, are the equivalent plastic strain rates and $f_i$ are other predefined field variables. In the case of unloading from any point on the strain-softening branch of the stress–strain curves, the unloading response exhibits lower stiffness (Figure 4), and the elastic stiffness of the material is degraded (damaged). The stiffness degradation is characterized by two damage variables, $d_t$ and $d_c$, that are considered to be functions of the plastic strains variables:

$$d_t = d_t\left(\bar{\varepsilon}_t^{pl}\right); \quad 0 \le d_t \le 1 \tag{7}$$

$$d_c = d_c\left(\bar{\varepsilon}_c^{pl}\right); \quad 0 \le d_c \le 1 \tag{8}$$

The damage variables are equal to 0 in the case of undamaged material and are equal to 1 in the case of a total loss of strength. If $E_0$ is the initial (undamaged) elastic stiffness of the material, the stress–strain relations under uniaxial tension and compression loading are, respectively:

$$\sigma_t = (1 - d_t)E_0\left(\varepsilon_t - \bar{\varepsilon}_t^{pl}\right) \tag{9}$$

$$\sigma_c = (1 - d_c)E_0\left(\varepsilon_c - \bar{\varepsilon}_c^{pl}\right) \tag{10}$$

Further, the "effective" tensile and compressive cohesion stresses are calculated as:

$$\bar{\sigma}_t = \frac{\sigma_t}{(1 - d_t)} = E_0\left(\varepsilon_t - \bar{\varepsilon}_t^{pl}\right) \tag{11}$$

$$\bar{\sigma}_c = \frac{\sigma_c}{(1 - d_c)} = E_0\left(\varepsilon_c - \bar{\varepsilon}_c^{pl}\right) \tag{12}$$

The effective cohesion stresses determine the yield (or failure) surface.

### 2.2.3. Variables of Damage and Stiffness Degradation

Complex degradation phenomena characterize concrete behavior under cyclic uniaxial loading that involves the re-opening and closure of previously developing micro-cracks along with their interaction. Based on experimental data, it has been verified that a part of the elastic stiffness is recovered as the loading sign changes during uniaxial testing. This stiffness recovery effect, also known as the "unilateral effect", is an important feature of concrete behavior under cyclic loading. It has been observed that the unilateral effect is pronounced as the load changes from tension to compression, causing the closure of the tensile cracks since it has; as a result, partial recovery of the compressive stiffness. The concrete damaged plasticity model assumes that the decrease in the elastic modulus can be given in terms of the scalar coefficient $d$; thus, the re-loading modulus of elasticity $E_o$ is reduced to $E$ as

$$E = (1 - d)E_0 \tag{13}$$

where $E_0$ is the initial modulus of elasticity of the undamaged concrete. This expression applies both in the tensile ($\sigma_{11} > 0$) and the compressive ($\sigma_{11} < 0$) direction of the cyclic loading. The stiffness degradation coefficient, $d$, is a function of the stress state and the uniaxial damage variables, $d_t$ and $d_c$. Lubliner et al. [31] support plastic degradation only

within the softening range, and stiffness depends on the material's cohesion. Thus, the plastic damage factor is assumed as

$$\frac{E}{E_o} = 1 - d = \frac{c}{c_{max}} \tag{14}$$

Additionally, thus $d = 1 - c/c_{max}$ where $c$ is cohesion proportion to stress and $c_{max}$ is proportional to concrete strength.

For the uniaxial cyclic conditions, it is assumed [20,32] that:

$$(1 - d) = (1 - s_t d_t)(1 - s_c d_c) \tag{15}$$

where $s_t$ and $s_c$ are functions of the stress state that are introduced to model stiffness recovery effects associated with stress reversals. They are defined according to the following:

$$s_t = 1 - w_t r^* \sigma_{11}; \quad 0 \le w_t \le 1 \tag{16}$$

$$s_c = 1 - w_c r^* \sigma_{11}; \quad 0 \le w_c \le 1 \tag{17}$$

where

$$r^*(\sigma_{11}) = \begin{cases} 1 & if \ \sigma_{11} > 0 \\ 0 & if \ \sigma_{11} < 0 \end{cases} \tag{18}$$

The weight factors $w_t$ and $w_c$, are considered to be material properties that control the recovery of the tensile and compressive stiffness upon load reversal. Parameter $r^*$ as defined in relationship (24), denotes that in compression stiffness recovery, the recovery is not reduced as it is in the case of the tensional stress state. Figure 5 presents this phenomenon in case the load changes from tension to compression.

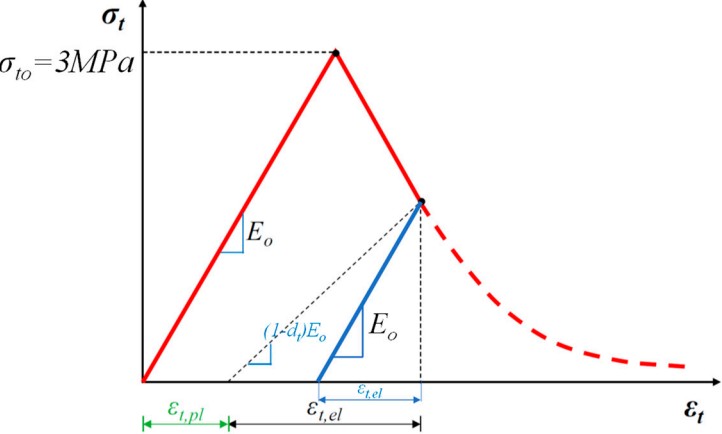

**Figure 5.** Effect of the compression stiffness recovery in case the load changes from tension to compression.

In case that there was no previous compressive damage in the material, that is;

$$(1 - d) = (1 - s_t d_t) = (1 - (1 - w_c(1 - r^*))d_t) \tag{19}$$

Then

$$\bar{\varepsilon}_c^{pl}(1 - d) = d_c = 0 \tag{20}$$

In tension ($\sigma_{11} > 0$), $r^* = 1$; therefore, $d = d_c$ as expected. In compression ($\sigma_{11} < 0$), $r^* = 0$ and $d = (1 - w_c)d_t$. If $w_c = 1$, then $d = 0$; therefore, the material fully recovers the compressive stiffness (and here is the initial undamaged stiffness, $E = E_0$) [20,30–34]. Effect of the compression stiffness recovery parameter $w_c$ is shown in Figure 6.

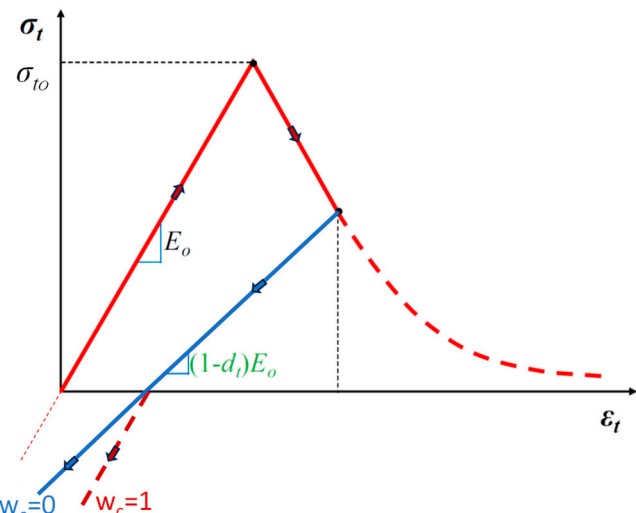

**Figure 6.** Effect of the compression stiffness recovery parameter $w_c$.

2.2.4. Postfailure Stress–strain Relation

In reinforced concrete, the specification of post-failure behavior is assumed that the post-failure stress is presented as a function of cracking strain $\bar{\varepsilon}_t^{ck}$. The cracking strain is defined as the total strain minus the elastic strain corresponding to the undamaged material; that is,

$$\bar{\varepsilon}_t^{ck} = \varepsilon_t - \varepsilon_{0t}^{el} \tag{21}$$

where $\varepsilon_{0t}^{el} = \sigma_t/E_o$, as illustrated in Figure 5.

The article discusses the Concrete Damaged Plasticity (CDP) model parameters utilized in the finite element analysis to accurately represent the behavior of reinforced concrete beam-column joints strengthened with CFRP ropes. The selection of these parameters was based on a comprehensive review of experimental data available in the literature, covering the used concrete mixtures, steel reinforcements, and FRP materials [29–35]. Key parameters such as the compressive and tensile strengths of concrete, the yield strength and modulus of elasticity of steel reinforcement, and the tensile strength and modulus of elasticity of CFRP ropes were carefully considered to ensure compatibility with the materials used in the study. Limited sensitivity analysis was also performed to evaluate the influence of variations in material properties on the model predictions, further refining the parameter values to enhance the predictive capability of the finite element analysis.

The compressive strength of the concrete used in the tested beam-column joints is based on complementary cylinder compression tests. Six standard cylinders with d = 150 mm and h = 300 mm were tested. The mean compression strength was almost 30 MPa. Consequently, the values used in the analyses were 30 MPa for maximum compression strength and 12.5 MPa for yielding (Figure 4b), whereas for maximum tension strength, 3.0 MPa.

Moreover, five parameters of the Concrete Damage Plasticity model have also to be defined for the analyses. These parameters are the dilatation angle ψ, the potential eccentricity $\varepsilon$, the ratio $f_{b0}/f_{c0}$, parameter K of the yielding surface, and the viscosity parameter. In particular:

— Dilatation angle characterizes the plastic deformation. Different values of this parameter are used in the literature [20,32]. A value equal to 56° leads to the ductile material response, which is not realistic for concrete, whereas a value close to 0 leads to an entirely brittle behavior. A value equal to 35 has been adopted for parameter ψ in the present study.

— Eccentricity $\varepsilon$ represents the rate of the deflection divergence of the plastic hyperbolic behavior to its asymptote. It is usually taken equal to the 0.10 value adopted in the present study, too.

- The value of the ratio of the biaxial strength $f_{b0}$ to the corresponding uniaxial strength $f_{c0}$ is equal to 1.16, as recommended by many researchers in the literature [20,32].
- Parameter K, which represents the ratio of the tensile meridian to the compressive one (Figure 3a), is usually recommended equal to 2/3 [20,32].
- FINALLY, the viscosity parameter is usually taken very small or equal to zero. A very small value equal to 0.00008 is adopted, which helps the analysis procedure reach good convergence [20,32–35].

In Table 2, the parameters of concrete elements used in the analysis of specimens are presented.

**Table 2.** Definition of concrete parameters in ABAQUS: (**a**) damage plasticity parameters, (**b**) concrete compressive behavior, and (**c**) concrete tensile behavior.

| (a) Concrete Damage Plasticity Parameters [20,32–35] | | (b) Concrete Compressive Behavior | | (c) Concrete Tensile Behavior | |
|---|---|---|---|---|---|
| | | Stress | Inelastic | Stress | Inelastic |
| | | | Strain | | Strain |
| Dilation Angle | 35 | 12.50 **(yield)** | 0.000000 | 3.0000 **(yield)** | 0.000000 |
| Eccentricity | 0.1 | 14.78 | $1.5 \times 10^{-5}$ | 1.66400 | 0.000281 |
| fb0/fc0 | 1.16 | 16.89 | $4.0 \times 10^{-5}$ | 1.78900 | 0.000507 |
| K | 0.667 | 18.81 | $8.0 \times 10^{-5}$ | 0.92300 | 0.000718 |
| Viscosity Parameter | 0.008% | 26.60 | 0.000130 | 0.76383 | 0.000923 |
| | | 28.60 | 0.000202 | 0.65420 | 0.001124 |
| | | 29.20 | 0.000300 | | |
| | | 30.00 | 0.000396 | | |

### 2.3. Steel Material and FRP Ropes

The quality of the used reinforcement steel was B500C. The mean tensile strength $f_y$ 550 MPa, whereas, for the analysis, the stress–strain relationship is considered to be elastic and perfectly plastic. The CFRP rope used for the strengthening of the specimens JA0Fxb and JA0F2x2b has tensile strength equal to 4000 MPa, modulus of elasticity equal to 240GPa and cross-section area $A_s > 28$ mm$^2$ (to manufacturer's data SikaWrap $^®$ FX-50 C, Sika Hellas SA, Kryoneri, Greece) [1,6].

While the study effectively demonstrates the immediate seismic response improvements achieved via the application of CFRP ropes for strengthening reinforced concrete beam-column joints, it is essential to consider the long-term durability and maintenance implications of this approach. CFRP materials have high strength and corrosion resistance, which can contribute to the longevity of strengthened structures. However, prolonged exposure to environmental conditions, including temperature variations, moisture ingress, and UV radiation, may affect the performance of CFRP ropes over time. To ensure the continued effectiveness of the strengthening solution, periodic inspections, and maintenance activities may be necessary. These could include visual assessments for signs of degradation, such as delamination or fiber exposure, as well as targeted repairs or replacements as needed. Additionally, research on the long-term durability of CFRP materials under cyclic loading and environmental exposure should be prioritized to provide further insights into their performance and inform best practices for maintenance and asset management in the field of structural rehabilitation.

### 3. Finite Element Simulation

In the performed analyses for the simulation of concrete, the 8-node 3-dimensional solid elements with reduced integration (C3D8R) are used [20] to avoid the effect of shear locking. Further, the 3-dimensional 2-node truss elements are used for the simulation of steel bars (longitudinal reinforcement and stirrups). Each element's node has three degrees

of freedom. The bond between reinforcement and concrete is modeled using the embedded process "truss in solid" [20]. Element meshes for the materials of the studied joints are presented in Figures 7–9.

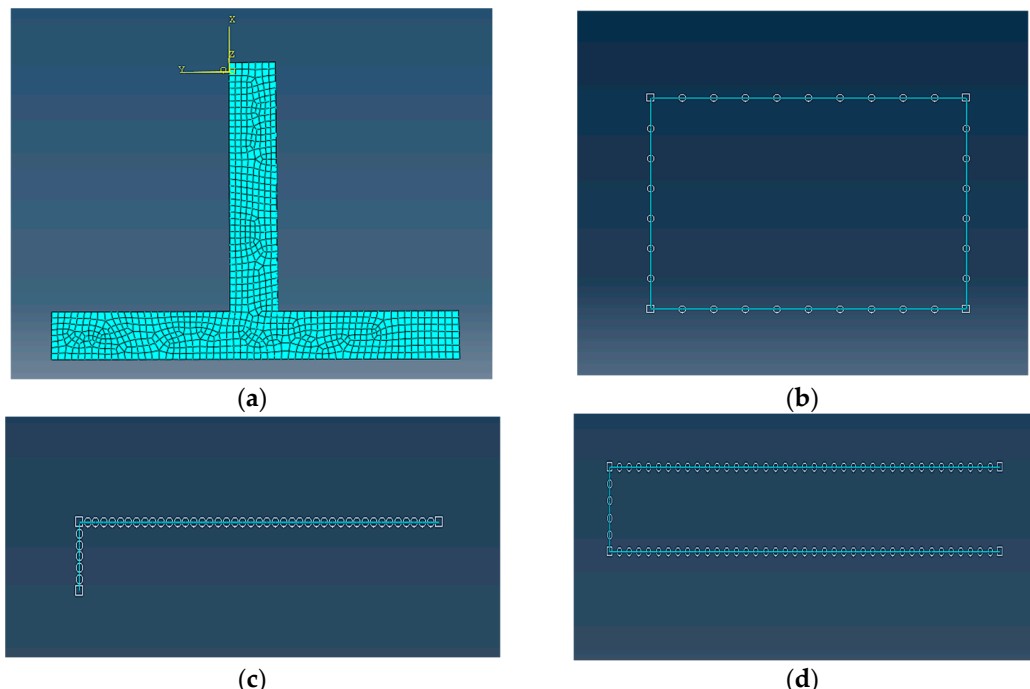

**Figure 7.** Finite element simulation of the parts of the specimen, (**a**) concrete elements, (**b**) stirrups, (**c**) steel bars, and (**d**) CFRP ropes.

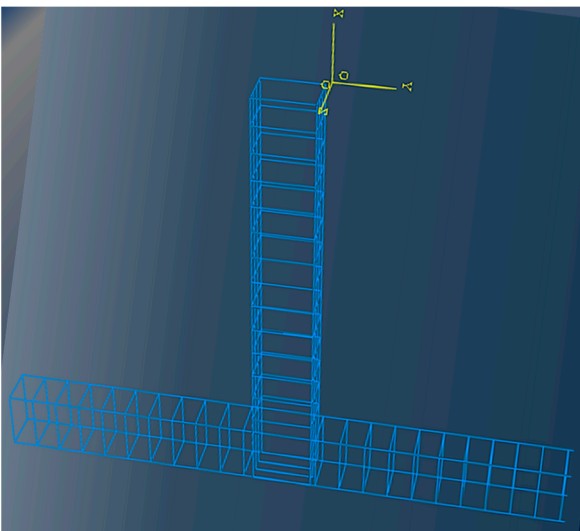

**Figure 8.** Steel reinforcement of the specimen.

Thereupon it is mentioned that while the finite element model employed in the study provides valuable insights into the behavior of reinforced concrete beam-column joints strengthened with CFRP ropes, it is important to recognize the inherent simplifications and assumptions made during the modeling process. One key assumption is the material behavior of concrete, steel reinforcement, and CFRP ropes, which are typically idealized using constitutive models such as the adopted ones in the Concrete Damaged Plasticity model. While these models capture the essential nonlinear behavior of the materials, they may not fully capture all complexities, such as strain-rate effects, creep, and environmental degradation, which could affect the long-term performance of the strengthened joints.

Additionally, the finite element analysis simplifies the geometric and boundary conditions of the specimens, assuming idealized loading and support conditions that may not fully replicate real-world scenarios. Furthermore, the bonding behavior between the CFRP ropes and concrete is simplified using the embedded truss model, neglecting potential interface debonding and slip effects under cyclic loading. These simplifications could impact the accuracy of the finite element analysis predictions, particularly in capturing the intricate interactions between material properties, geometry, and loading conditions.

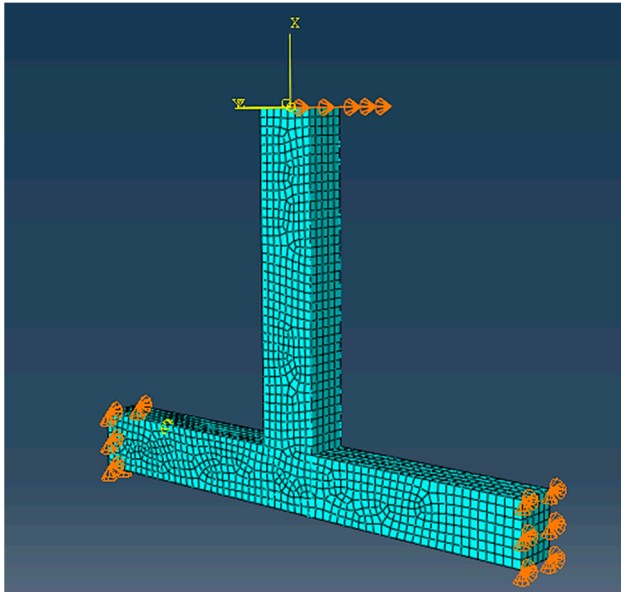

**Figure 9.** Boundary conditions of the specimen and mesh.

A sensitivity analysis was conducted to assess the robustness of the finite element model predictions to variations in material properties of concrete, steel reinforcement, and CFRP ropes. Different scenarios were considered by varying the Young's modulus, yield strength, and ultimate tensile strength of these materials within a reasonable range of values. The analysis revealed that the model predictions were sensitive to changes in material properties, particularly for parameters such as the maximum load-bearing capacity, shear deformation, and crack propagation patterns. Specifically, variations in the modulus of elasticity of concrete and CFRP ropes resulted in noticeable differences in the stiffness and overall behavior of the strengthened joints under seismic loading. Similarly, changes in the yield strength of steel reinforcement significantly influenced the onset and propagation of plastic deformations within the joints. These findings highlight the importance of accurately characterizing material properties in the finite element model to ensure reliable predictions of the structural response. Additionally, the sensitivity analysis underscores the need for thorough material testing and calibration procedures to enhance the robustness of the proposed strengthening method and improve its applicability to a wide range of structural configurations and loading conditions.

### 3.1. Loading, Mesh and Convergence

The test setup, along with a presentation of the measuring instrumentation, is presented in Figure 10a. The examined beam-column connections are rotated 90o and located with the column in the horizontal way, whereas the beam is in the vertical direction. The column, which is the horizontal part of the specimen, is supported at its ends using devices that allow rotation to idealize the inflection points in the middle height of columns in multi-story reinforced concrete frame structures. An axial compressive load equal to 0.05 Acfcm was applied to the column (horizontal element) during the test.

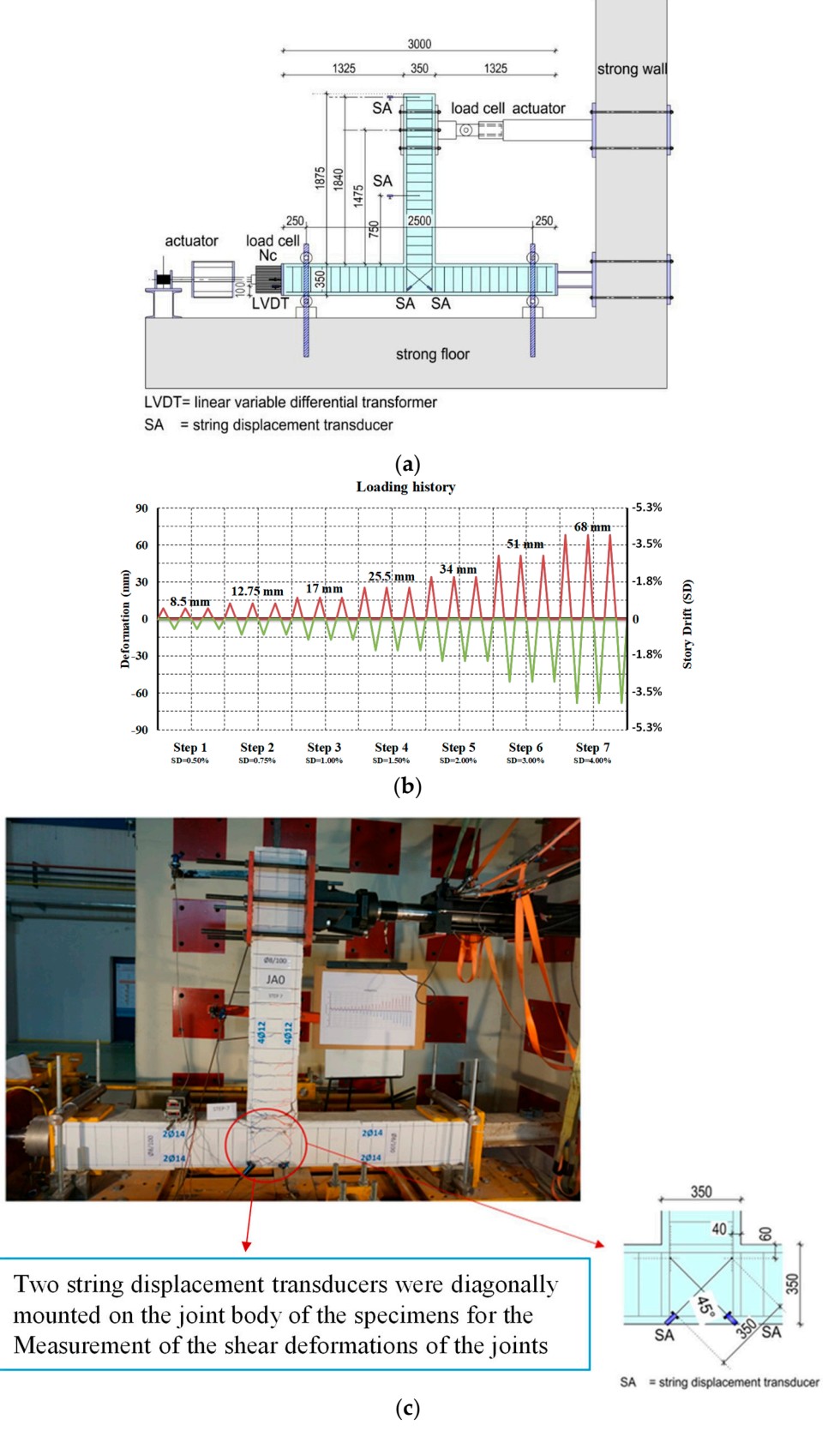

**Figure 10.** Test setup, loading sequence, and instrumentation for the measurement of shear deformation of the joint body of the tested specimens. (**a**) Experimental setup. (**b**) Loading sequence. (**c**) Specimen JA0 at the end of the test.

Specimens were subjected to cyclic loading imposed as cyclic movements of the free end of the beam (vertical element) by an actuator. The tested joints JA0, JA1, JA0Fxb, and JA0F2x2b were subjected to seven loading steps (Figure 10b), whereas each loading step included three full loading cycles. The maximum displacements of the free end of the beam at each loading step were ±8.5 mm, ±12.75 mm, ±17.0 mm, ±25.5, ±34.0, ±51.0, ±68.0 mm (Figure 10b). Figure 10c presents specimen JA0 at the end of the test.

At the beginning of the loading procedure, as mentioned before, an axial load was applied to the column of the joint specimen. Thenceforth, displacement was imposed at the end point of the beam, as presented in Figure 10. The displacement was transversely imposed in a quasi-static manner in both directions, upwards and downwards, since the loading was a cyclic one. It is stressed that the displacement was applied smoothly, keeping a constant rhythm in order to achieve a quasi-static solution and prevent any essential acceleration alteration through each iteration.

Mesh size was chosen considering that the distribution of concrete cracking typically includes spatial scales between two to three dominant aggregate sizes of the concrete mixture. The aggregate size for the specimens was 16 mm.

Moreover, the mesh size of concrete should be close to the values of typical concrete cubes, which are usually used for the estimation of concrete compressive strength. Thus, mesh sizes of 50 mm for steel elements and FRP elements and 70 mm for concrete elements have been selected [17,33,34].

*3.2. Boundary Conditions*

The beam-column joint specimen is considered to be pinned on the top, whereas it is considered to be with a moving pin at the bottom of the column in order to receive the constant axial load. It is necessary because the bending moment in the middle of the deformable height of a real column of a frame is approximately zero for horizontal seismic loads. Thus, the supports have to allow the rotation of the two boundaries of the column part of the specimen. The boundary conditions and specimen finite element discretization can be seen in Figure 9. The loading sequence is applied as imposed displacements at the end of the beam (Figure 10).

## 4. Test Setup and Measurement of Shear Deformations

The test setup of the applied loading sequence and a photograph of the specimen JA0 are shown in Figure 10. Displacements of the joint specimen are given as Story Drifts (SD). In particular, as drift is implied, the ratio of the imposed displacement to the length of the beam is measured from the loaded end until the column centerline. In the tested joints, drift is calculated based on the observed deformation $\Delta l$ at each loading step as follows [6]:

$$\Delta \ell / (\ell_b + \frac{h_c}{2}) = \Delta \ell / (1.525 + \frac{0.35}{2}) = \Delta \ell / 1700 \text{ mm}$$

Moreover, in the experimental procedure, it is important for the shear deformations of the joint body of the beam-column connections to be measured. It can be achieved using two string LVDTs externally mounted on the joint panel. These LVDTs are diagonally placed and have the capacity to record the elongation and the shortening of the diagonals of the joint panel at each step of the loading, as can be seen in Figure 10 [6].

## 5. Numerical Results-Comparison with Experimental

Herein, the results of the experiments and the FE analysis of the four specimens are presented. Numerical cracking patterns, principal stresses, shear deformation, and force displacement curves are thoroughly discussed. As we can see, the differences are small, and it proves that the analysis is accurate enough.

*5.1. Numerical Results and Cracking Patterns*

In Figure 11, comparisons between the observed from the tests and the numerical cracking patterns after the final loading step (step 7) are presented. In particular, specimen JA0 in Figure 11a,b, specimen JA1 in Figure 11c,d, strengthened specimen JA0Fxb in Figure 11e,f, and finally, the strengthened specimen JA0F2x2b in Figure 11g,h.

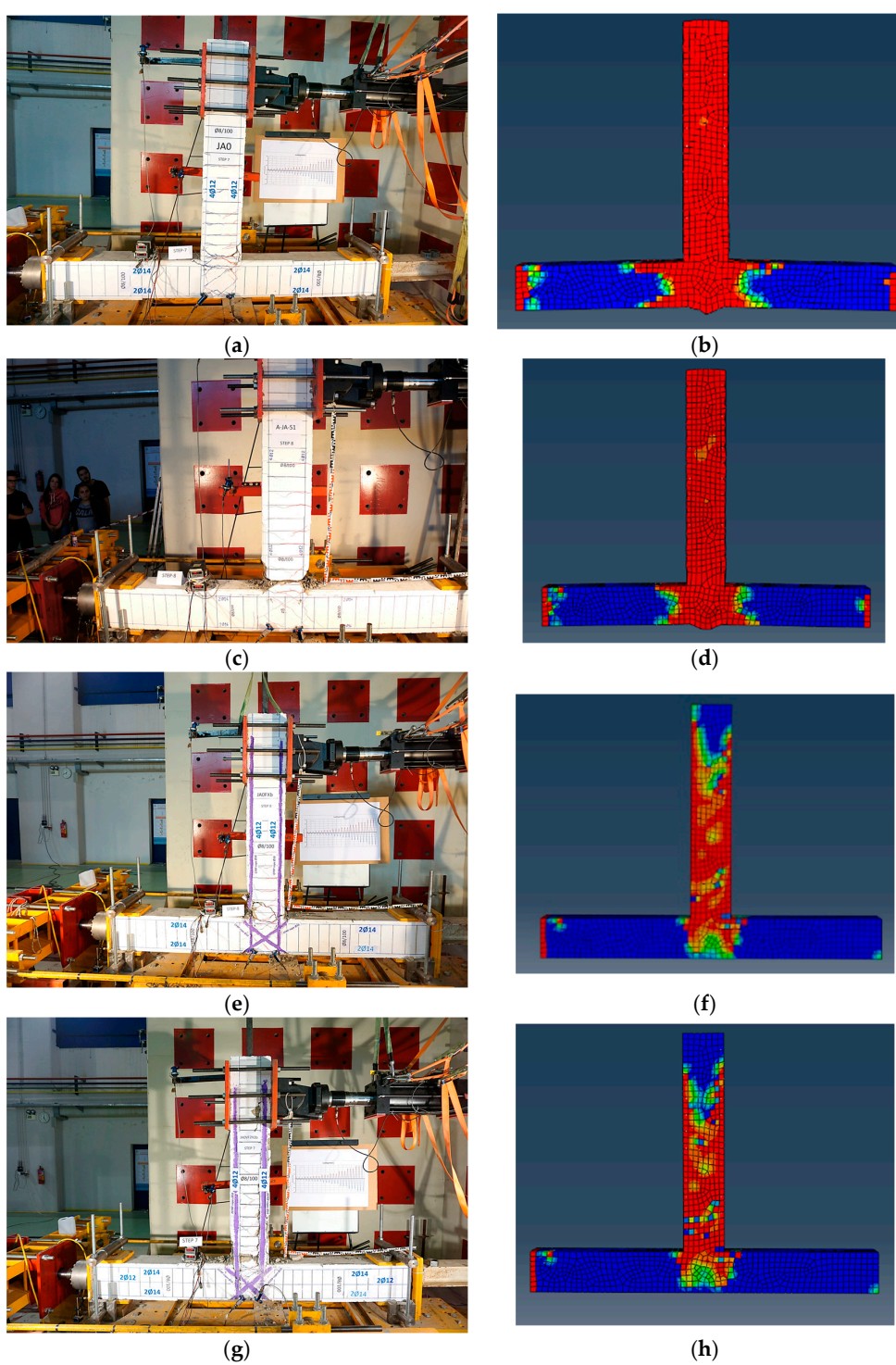

**Figure 11.** Comparison between Experimental and Numerical cracking patterns, in the final step (step 7), (**a**,**b**) specimen JA0, (**c**,**d**) specimen JA1, (**e**,**f**) specimen JA0Fxb, (**g**,**h**) specimen JA0F2x2b.

*5.2. Comparison of Numerical Results with Experimental Ones*

Results, as yielded from the performed analyses, are compared with the corresponding experimental ones in order to assess the validity and accuracy of the attempted approach. In this direction, maximum principal stresses, force–displacement curves, and shear deformations in the joint body are presented for all specimens, and the numerical results are compared with the experimentally acquired data. In particular, Figures 12–14 presented these comparisons for specimen JA0, Figures 15–17 for specimen JA1, and Figures 18–20 present comparisons for the strengthened specimen JA0Fxb, and finally, Figures 21–23 for the strengthened specimen JA0F2x2b. In particular, the following remarks can be drawn.

5.2.1. Pilot Specimens JA0 and JA1

— Specimen JA0. Experimental and numerical results of the principal stresses developing in the joint body of the specimen are presented in Figure 12a–c for the 1st, 2nd, and 3rd loading cycles of the loading steps, respectively. Red dashed lines represent the observed values, whereas blue lines represent the numerical results. From these comparisons, it is apparent that the numerical approach successfully calculates the principal stresses in the joint body. Further, Figure 13a–c presents the numerical values (blue lines) versus the experimentally measured values (red dashed lines) of the maximum displacements at each loading step for the 1st, 2nd, and 3rd loading cycles of the loading steps, respectively. From the comparisons, it is concluded that the numerical approach, in general, successfully describes the experimental ones. Perhaps a small discrepancy appears in the results of the negative loadings in the maximum loadings of the 2nd cycles under large loading, perhaps due to the experimental measurements. Finally, joint shear deformations of the joint body presented in Figure 14 show discrepancies between experimental and numerical values between experimental and numerical values in the middle part of the loading steps, perhaps due to the experimental measurements during the test procedure.

— Specimen JA1. Experimental and numerical results of the principal stresses developing in the joint body of the specimen are presented in Figure 15a–c for the 1st, 2nd, and 3rd loading cycles of the loading steps, respectively. Red dashed lines represent the observed values, whereas blue lines represent the numerical results. From these comparisons, it is apparent that the numerical approach successfully calculates the principal stresses in the joint body. Further, Figure 16a–c presents the numerical values (blue lines) versus the experimentally measured values (red dashed lines) of the maximum displacements at each loading step for the 1st, 2nd, and 3rd loading cycles of the loading steps, respectively. From the comparisons, it is concluded that the numerical approach successfully describes the experimental ones. Finally, from joint shear deformations of the joint body presented in Figure 17, it can be concluded that numerical results successfully depict the tendency and are very close to the measured values obtained from the experiment.

5.2.2. Strengthened Specimens JA0Fxb and FA0F2x2b

— Specimen JA0Fxb. Experimental and numerical results of the principal stresses developing in the joint body of the specimen are presented in Figure 18a–c for the 1st, 2nd, and 3rd loading cycles of the loading steps, respectively. Red dashed lines represent the observed values, whereas blue lines represent the numerical results. From these comparisons, it is apparent that the numerical approach excellently calculates the principal stresses in the joint body. Further, Figure 19a–c presents the numerical values (blue lines) versus the experimentally measured values (red dashed lines) of the maximum displacements at each loading step for the 1st, 2nd, and 3rd loading cycles of the loading steps, respectively. From the comparisons, it is concluded that the numerical approach successfully describes the experimental ones. Finally, joint shear deformations of the joint body presented in Figure 17 show that numerical results successfully predict the measured shear deformations.

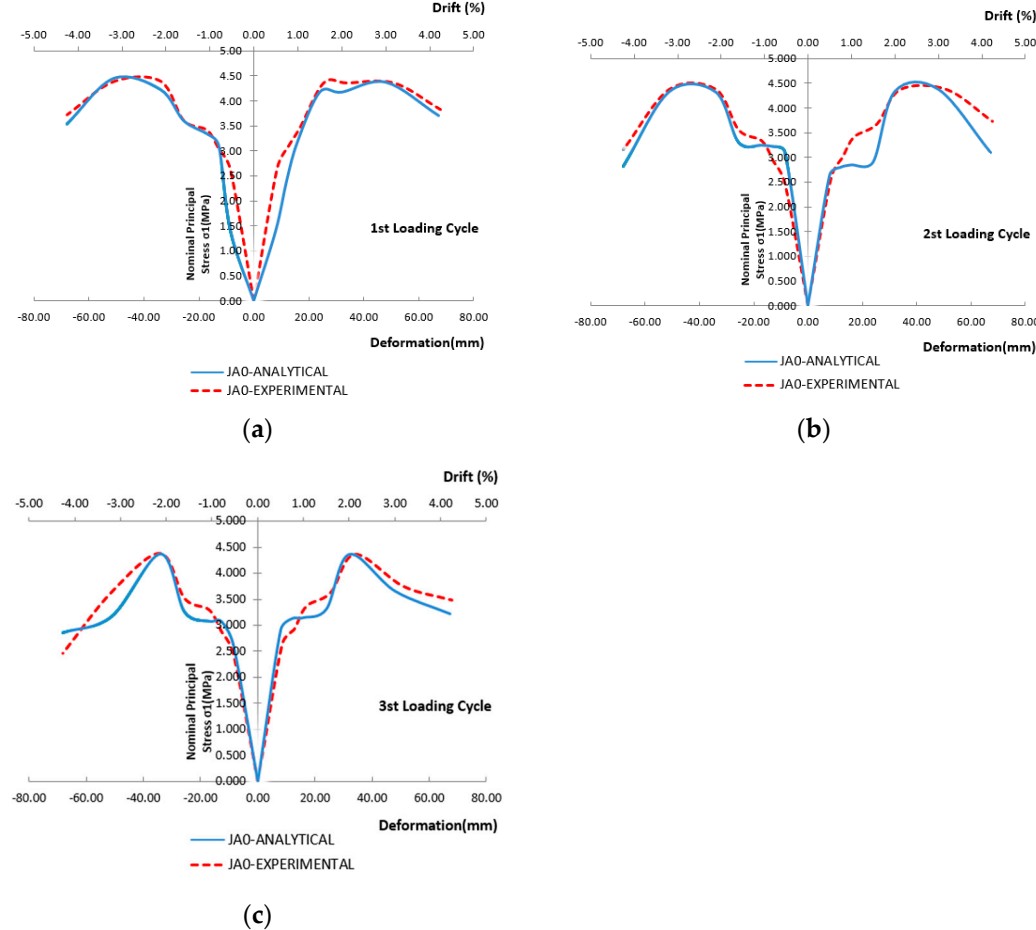

**Figure 12.** Experimental and Numerical results of principal stresses developing in the joint body of specimen JA0, (**a**) for 1st loading cycles, (**b**) for 2nd loading cycles, (**c**) for 3rd loading cycles.

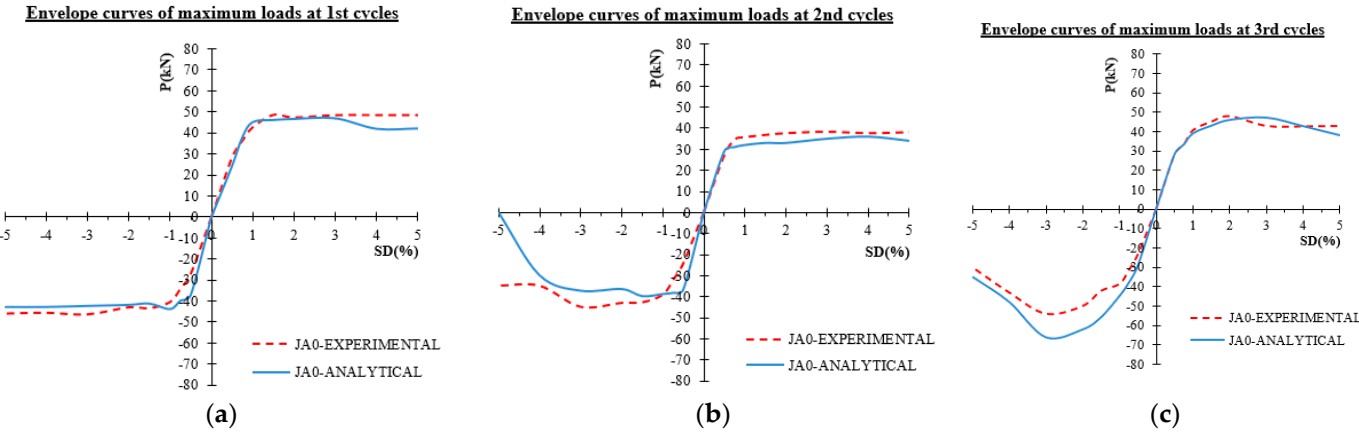

**Figure 13.** Experimental and Numerical results of force displacement of specimen JA0, (**a**) for 1st loading cycles, (**b**) for 2nd loading cycles, (**c**) for 3rd loading cycles.

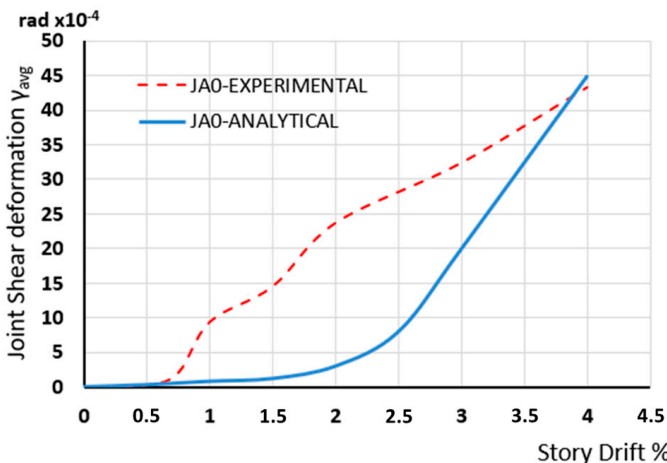

**Figure 14.** Experimental and Numerical results of shear deformation developing in the middle of joint body of specimen JA0.

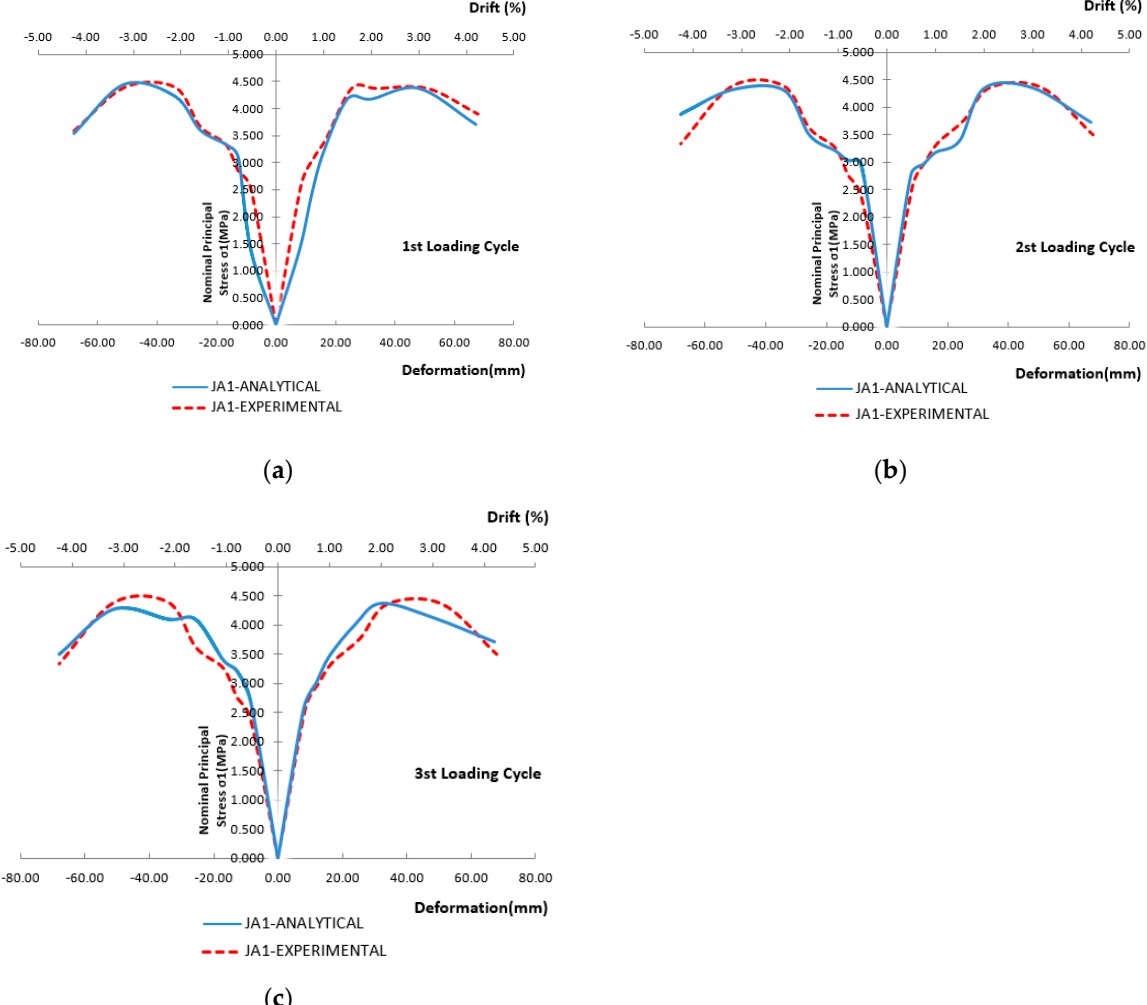

**Figure 15.** Experimental and Numerical results of maximum principal stress of specimen JA1, (**a**) for 1st loading cycles, (**b**) for 2nd loading cycles, (**c**) for 3rd loading cycles.

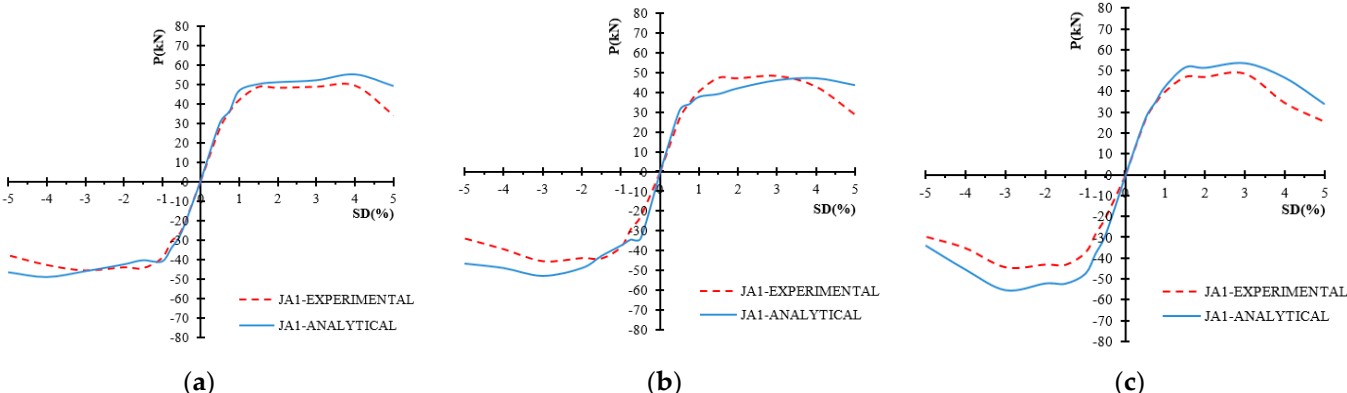

**Figure 16.** Experimental and Numerical results of force displacement of specimen JA1, (**a**) for 1st loading cycles, (**b**) for 2nd loading cycles, (**c**) for 3rd loading cycles.

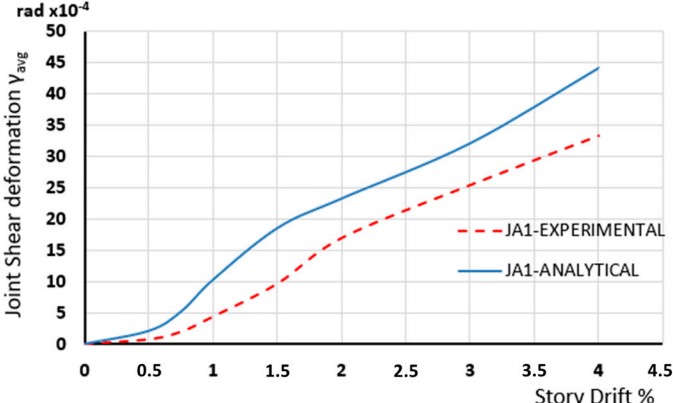

**Figure 17.** Experimental and Numerical results of shear deformation developing in the joint body of specimen JA1.

— Specimen JA0F2x2b. Experimental and numerical results of the principal stresses developing in the joint body of the specimen are presented in Figure 21a–c for the 1st, 2nd, and 3rd loading cycles of the loading steps, respectively. Red dashed lines represent the observed values, whereas blue lines represent the numerical results. From these comparisons, it is apparent that the numerical approach excellently calculates the principal stresses in the joint body. Further, Figure 22a–c presents the numerical values (blue lines) versus the experimentally measured values (red dashed lines) of the maximum displacements at each loading step for the 1st, 2nd, and 3rd loading cycles of the loading steps, respectively. From the comparisons, it is concluded that the numerical approach successfully describes the experimental ones. Finally, joint shear deformations of the joint body are presented in Figure 23. From these comparisons, it is shown that numerical results successfully depict the tendency and are very close to the measured values obtained from the experiment. Discrepancies shown in high-story drifts may be attributed to the measurements of the damaged joint body.

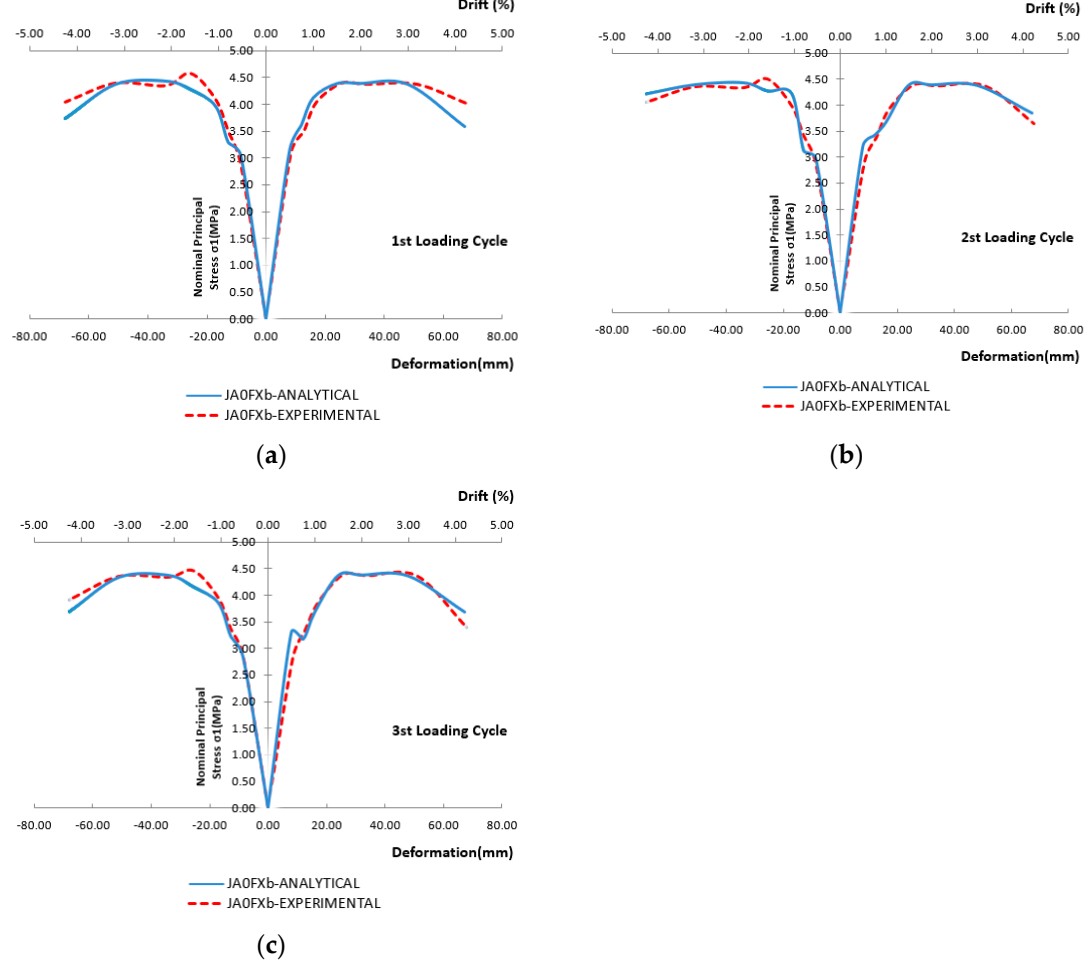

**Figure 18.** Experimental and Numerical results of maximum principal stress of specimen JA0Fxb, (**a**) for 1st loading cycles, (**b**) for 2nd loading cycles, (**c**) for 3rd loading cycles.

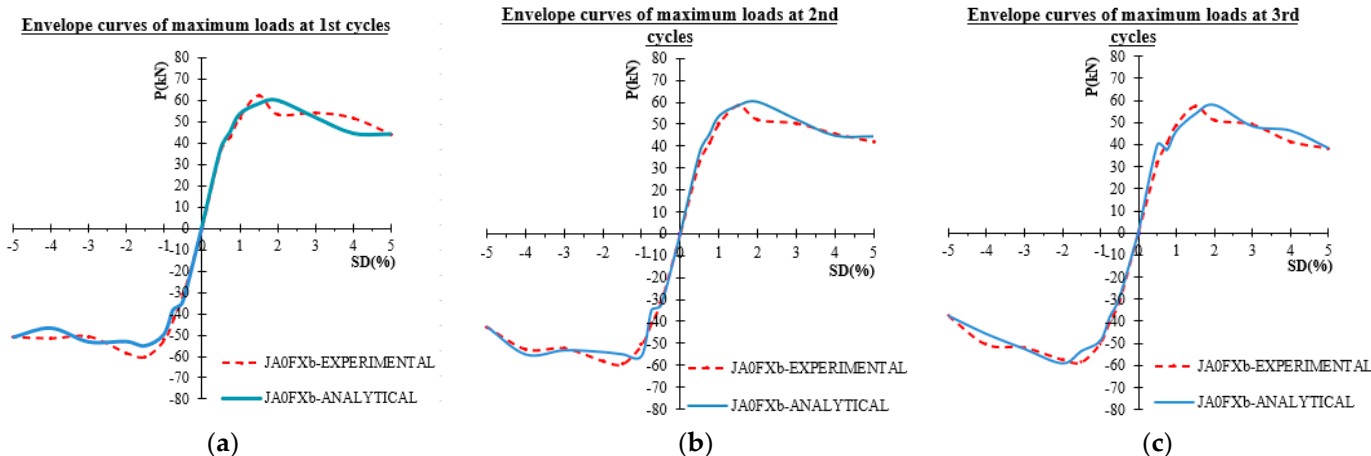

**Figure 19.** Experimental and Numerical results of force displacement of specimen JA0Fxb, (**a**) for 1st loading cycles, (**b**) for 2nd loading cycles, (**c**) for 3rd loading cycles.

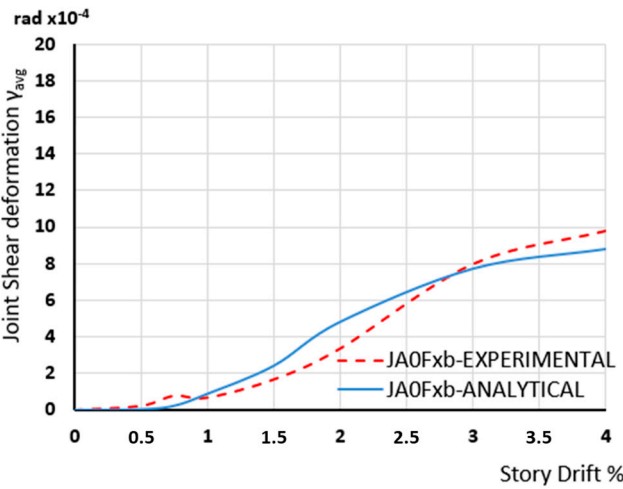

**Figure 20.** Experimental and Numerical results of shear deformation developing in the joint body of specimen JA0Fxb.

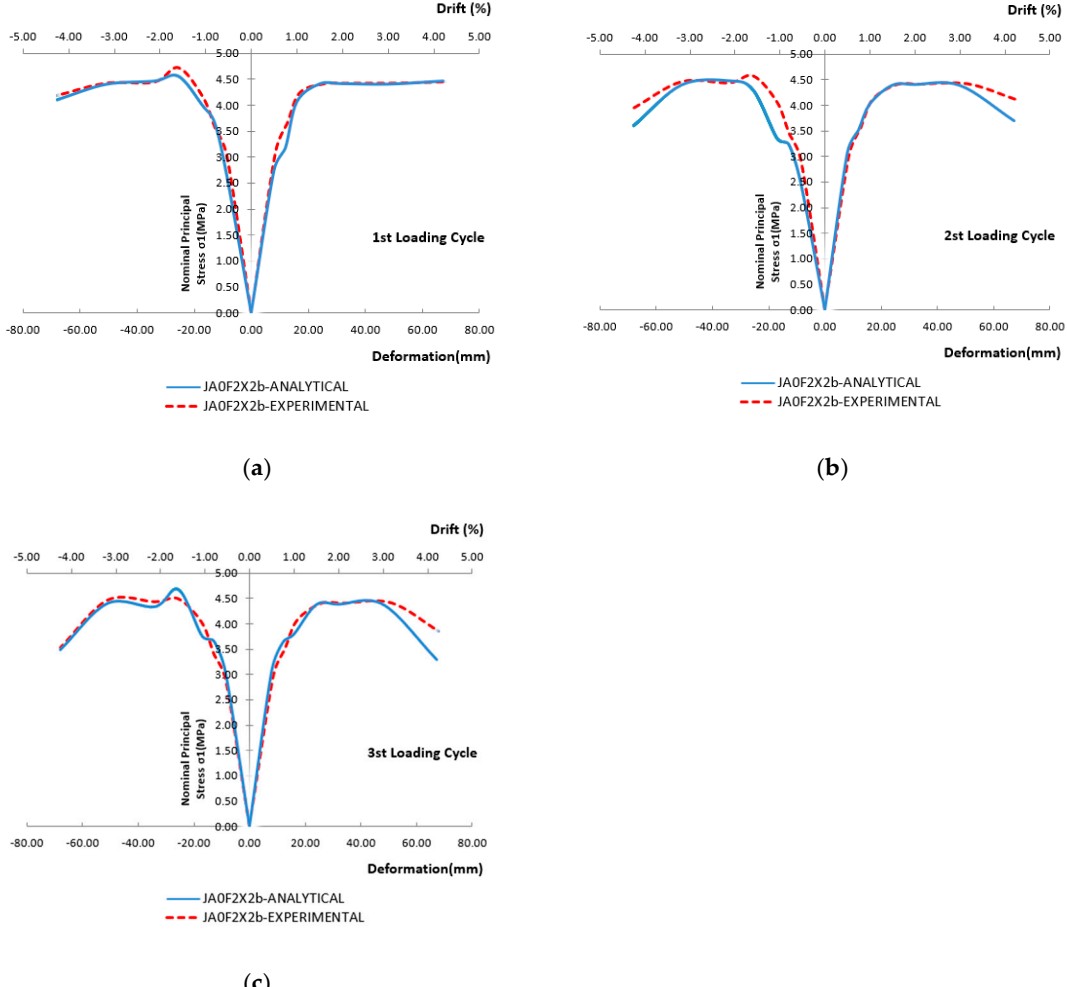

**Figure 21.** Experimental and Numerical results of maximum principal stress of specimen JA0F2x2b, (**a**) for 1st loading cycles, (**b**) for 2nd loading cycles, (**c**) for 3rd loading cycles.

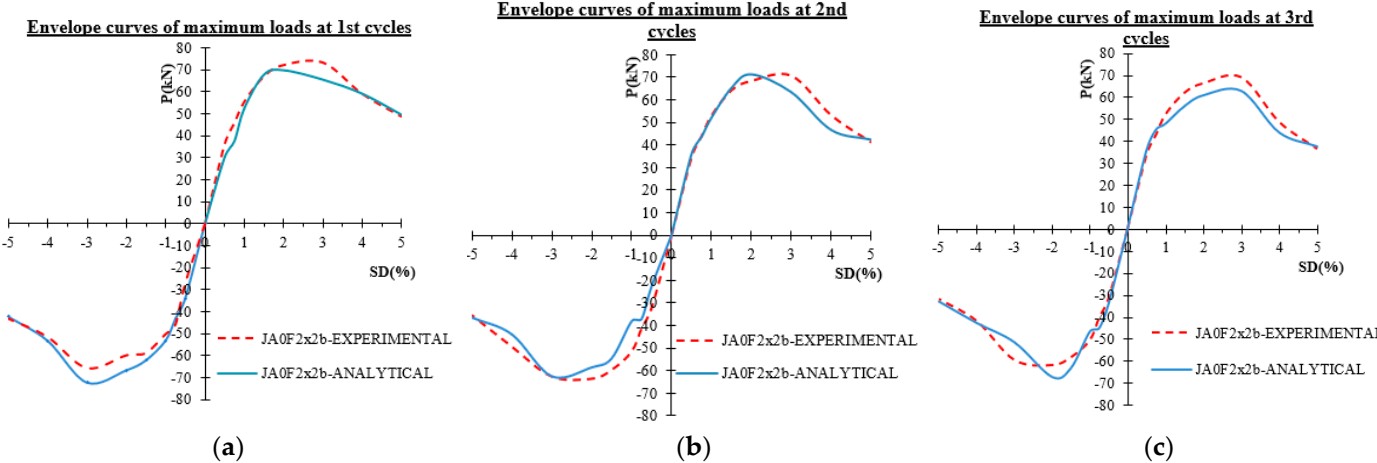

**Figure 22.** Experimental and Numerical results of force displacement of specimen JA0F2x2b, (**a**) for 1st loading cycles, (**b**) for 2nd loading cycles, (**c**) for 3rd loading cycles.

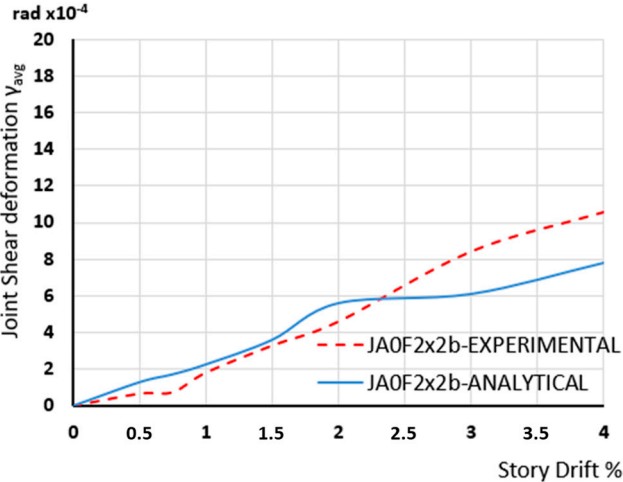

**Figure 23.** Experimental and Numerical results of shear stresses developing in the joint body of specimen JA0F2x2b.

## 6. Comparisons of Numerical Results with the Experimental Ones—Discussion

The numerical results show that the adopted analysis using 3D finite elements can predict the behavior of the examined specimens well. It is very important that the big diagonal cracks in the joint body are developed when the biggest principal stress is achieved. In Figures 12, 15, 18 and 21, the principal stresses of specimens JA0, JA1, JA0Fxb, and JA0F2x2b, respectively, as yielded using the numerical approach, are presented and compared with the corresponding ones as observed during the tests for the 1st, the 2nd and the 3rd loading cycles. From these comparisons, it is concluded that the adopted approach successfully predicts the development of principal stresses in the joint bodies of the examined beam-column specimens.

Further, in Figures 13, 16, 19 and 22, the envelope curves of the maximum loads of the hysteretic responses of specimens JA0, JA1, JA0Fxb, and JA0F2x2b, respectively, as yielded using the numerical approach are presented and compared with the corresponding ones as observed during the tests for the 1st, the 2nd and the 3rd loading cycles. From these comparisons, it is concluded that the adopted approach successfully predicts the development of maximum load per loading cycle for all loading steps of the examined beam-column specimens.

Finally, in Figures 14, 17, 20 and 23, the shear deformations $\gamma_{avg}$ of specimens JA0, JA1, JA0Fxb, and JA0F2x2b, respectively, as yielded using the numerical approach are presented and compared with the corresponding ones as calculated based on the observed deformations of the string displacement transducers mounted on the joints body (Figure 10) during the tests [6]. From these comparisons, it is concluded that the adopted approach adequately predicts the developing shear deformations per loading step of the examined beam-column specimens in most cases.

It is emphasized that the comparison between numerical predictions and experimental observations reveals that the adopted finite element analysis method accurately predicts the behavior of the examined specimens. Notably, significant diagonal cracks in the joint body develop when the highest principal stress is reached, affirming the reliability of the numerical approach.

Figures depicting principal stresses, maximum loads of hysteretic responses, and shear deformations for different loading cycles are presented for each specimen. These comparisons demonstrate that the numerical approach successfully predicts the evolving mechanical responses of the beam-column specimens throughout the loading process.

Furthermore, it can also be concluded that the use of the innovative CFRP ropes as external reinforcement diagonally placed in superficial notches has been proven an efficient and easy-to-apply technique for the strengthening of substandard reinforced concrete beam-column joints [1–6,30,31].

## 7. Conclusions

The efficiency of the Finite Element Method for reinforced concrete beam-to-column joints has been investigated in Abaqus. The joints were either retrofitted with CFRP ropes or not. Based on the results of the study, the following remarks can be drawn:

- The differences between the Experimental and Numerical results are small, considering the load–displacement curves, the maximum principal stress, and the shear deformations. It shows that the material input in the program and the FE modeling have been accurately done.
- The only considerable difference was found in the shear deformation of specimen JA0.
- Further, the importance and effectiveness of the application of FRP ropes for the improvement of the seismic response of the joints have also been proved.
- The cracking patterns of the examined specimens, as predicted using the finite elements in specimens, are very close to those of the experimental ones. This shows that the used CDP can accurately predict the crack propagation in concrete, and it can simulate the concrete's triaxial behavior accurately.
- The favorable influence of the ropes is evaluated based on their real characteristics in order to achieve a more realistic prediction of SFRC behavior under compression and tension. The cyclic loading tests of retrofitted joints exhibit improved hysteretic responses in terms of stiffness, load-bearing capacity, deformation, and cracking behavior.

The developed nonlinear FE analysis accurately predicts the overall hysteretic response of joints and the beneficial effect of the added FRP ropes. Comparisons between the test and numerical results reveal that the developed nonlinear FE analysis with a smeared crack model that takes into account the tension softening and residual stiffness effect accurately predicts the hysteretic response of realistic concrete joints with steel reinforcement.

**Author Contributions:** Conceptualization, E.G.; methodology, P.T. and E.G.; software, P.T.; validation, C.G.K.; formal analysis, E.G., C.G.K. and P.T.; investigation, E.G., C.G.K. and P.T.; resources, E.G. and P.T.; data curation, E.G. and C.G.K.; writing—original draft preparation, C.G.K. and P.T.; writing—review and editing, C.G.K. and P.T.; visualization, P.T.; supervision, E.G.; project administration, C.G.K. and E.G. All authors have read and agreed to the published version of the manuscript.

**Funding:** This research received no external funding.

**Institutional Review Board Statement:** Not applicable.

**Data Availability Statement:** Available by authors.

**Acknowledgments:** Authors acknowledge that the materials have been provided by SIKA HEL-LAS SA.

**Conflicts of Interest:** The authors declare no conflicts of interest.

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
