# Peer review of "Seismic Response of RC Beam-Column Joints Strengthened with FRP ROPES, Using 3D Finite Element: Verification with Real Scale Tests"

_2673-4109, doi:10.3390/civileng5020020_

Round 1
Reviewer 1 Report
Comments and Suggestions for Authors
The paper titled "Seismic response of RC beam-column joints strengthened with 2 FRP ropes, using 3D finite element. Verification with real scale 3 tests", this titled not require a dot at the end.
Here is some comments on the paper
Abstract
It is required to have more results should be included in this abstract.
Introduction
This section is too short and should include all details on the finite element and numerical analysis with simulation via ABAQUS such as "Experimental and numerical analysis of 3D printed cement mortar specimens using inkjet 3DP".
It also gives some differences of abaqus compared with other software in general.
Materials and Methods
This section can have more details on materials regarding physical and chemical properties.
In the method in terms of the data and boundary condition which information has been included, please give all in detail.
Results
This part is okay.
Discussion
The author can expand and discuss more the differences between simulation and real-life tests and five may a model (real-life application) to have in future work.
Conclusion
Please give outcomes in a few bullet points.
References
Include necessary references
Authors should fix the similarities and reduce them to a maximum of 10%.
Comments on the Quality of English Language
overall is good, only proofreading is required.
Author Response
Reviewer 1:
The authors sincerely thank the reviewer for his appreciation and good advices. The responses to each of them are detailed below, and the revisions are indicated using red color in the revised manuscript.
The paper titled "Seismic response of RC beam-column joints strengthened with 2 FRP ropes, using 3D finite element. Verification with real scale 3 tests", this titled not require a dot at the end.
It has been amended.
Here is some comments on the paper
Abstract
It is required to have more results should be included in this abstract.
Thank you for the comment. More comments and results have been included in the abstract.
“In particular, comparisons between experimental and numerical results of principal stresses developing in the joint body of all examined specimens along with similar comparisons of force-displacement envelopes and shear deformation of the joint body confirmed the adequacy of the applied finite element approach for the investigation of the use of CFRP-ropes as an efficient and easy-to-apply strengthening technique.”
Introduction
This section is too short and should include all details on the finite element and numerical analysis with simulation via ABAQUS such as "Experimental and numerical analysis of 3D printed cement mortar specimens using inkjet 3DP".
Thank you for the comment. More details and comments have been applied in the introduction.
“The stress-strain relationship for steel was modelled using an elastic-perfectly plastic model, while the behavior of carbon fiber reinforced polymer (C-FRP) ropes, utilized for reinforcement, was represented with a linear elastic model due to their limited capacity for elastic tension. This comprehensive modelling approach allowed for the accurate simulation of the unique response of reinforced concrete elements under cyclic loading conditions.”
“Numerical simulations offer a cost-effective means to study complex structural behavior. ABAQUS, a widely-used finite element analysis software, provides robust capabilities for modelling concrete and steel behavior under various loading conditions; recent literature indicated the efficiency of finite elements and numerical analysis with simulation via ABACUS (Shakor et al [30]).”
It also gives some differences of ABACUS compared with other software in general.
The following comment has also been added
“It is worth noting that ABAQUS offers distinct advantages over other finite element software due to its robust capabilities for modeling nonlinear material behavior, complex geometries, and dynamic loading conditions.”
Furthermore, the following comments have also been added in the introduction:
“The geometrical characteristics of the specimens were meticulously designed to resemble common structures encountered in real-world applications. This design approach aimed to ensure the relevance and applicability of the experimental findings to a broader spectrum of structural configurations typically found in practice. As mentioned before the dimensions, the reinforcements and the overall design of these specimens comply with the corresponding ones of external beam-column connections of a multi-story reinforced concrete frame structure. Therefore, it is concluded that limitations can be considered only regarding the dimensions of the tested elements. Based on these considerations and given the limited existing research in this area the examined strengthening technique with the use of CFRP ropes, has to be further investigated experimentally and numerically based on tested specimens with substantially larger dimensions. However, while the specimens were designed to emulate common structural configurations, it is essential to acknowledge that the study's findings may have limitations in fully capturing the complexity and variability inherent in real-world structures. Factors such as variations in material properties, construction practices, and environmental conditions could influence the performance of actual structures differently than observed in controlled laboratory settings. Therefore, while the specimens offer valuable insights, their representativeness for a wider range of real-world applications should be considered within the context of specific structural designs and environmental conditions."
Materials and Methods
This section can have more details on materials regarding physical and chemical properties.
Thank you for the constructive comment. More details on materials regarding physical and other properties have been included in this paragraph (paragraph 2.1).
“Carbon fiber-reinforced polymer (CFRP) ropes are composite materials composed of high-strength carbon fibers embedded in a polymer matrix, typically epoxy resin. The physical properties of CFRP ropes are characterized by their high tensile strength, stiffness, and low weight, making them ideal for structural reinforcement applications. The carbon fibers provide excellent mechanical properties, with tensile strengths ranging from 2000 MPa to 7000 MPa, and modulus of elasticity ranging from 200 GPa to 800 GPa, depending on the manufacturing process and fiber orientation; the mechanical characteristics of the CFRP ropes used in this study as given by the manufacturer are: tensile strength 4000 MPa, tensional modulus of elasticity 240 GPa and cross-section of Carbon fibers Af=28 mm2. CFRP ropes also exhibit excellent corrosion resistance and durability, making them suitable for application in harsh environmental conditions. Additionally, CFRP ropes can be easily fabricated into various shapes and configurations, allowing for flexible application in strengthening structural elements such as beam-column joints in reinforced concrete structures. The chemical properties of CFRP ropes are primarily determined by the epoxy resin matrix, which provides adhesion between the carbon fibers and protects them from environmental degradation. Epoxy resins offer high chemical resistance, low shrinkage during curing, and excellent bonding properties, ensuring long-term performance and durability of CFRP strengthening systems.”
Results
This part is okay.
Discussion
The author can expand and discuss more the differences between simulation and real-life tests and five may a model (real-life application) to have in future work.
In Paragraph 6 more comments and discussion have been included in the revised manuscript.
“It is emphasized that the comparison between numerical predictions and experimental observations reveals that the adopted finite element analysis method accurately predicts the behaviour of the examined specimens. Notably, significant diagonal cracks in the joint body develop when the highest principal stress is reached, affirming the reliability of the analytical approach.”
“Figures depicting principal stresses, maximum loads of hysteretic responses, and shear deformations for different loading cycles are presented for each specimen. These comparisons demonstrate that the numerical approach successfully predicts the evolving mechanical responses of the beam-column specimens throughout the loading process.”
Conclusion
Please give outcomes in a few bullet points.
Following the reviewer’s comments outcomes are presented in bullet points.
References
Include necessary references
More necessary references have been added:
- Shakor, P.; Gowripalan N.; Rasouli H. Experimental and numerical analysis of 3D printed cement mortar specimens using inkjet 3DP. Archives of Civil and Mechanical Engineering 2021, 21:58. https://doi.org/10.1007/s43452-021-00209-3
- Genikomsou, A.S.; Polak, M.A. Finite element analysis of punching shear of concrete slabs using damaged plasticity model in ABAQUS. Eng. Struct. 2015, 98, 38–48. https://DOI.org/10.1016/j.engstruct.2017.01.024
- Hany, N.F.; Hantouche, E.G.; Harajli, M.H. Finite element modeling of FRP-confined concrete using modified concrete damaged plasticity. Eng. Struct. 2016, 125, 1–14.
- Chi, Y.; Yu, M.; Huang, L.; Xu, L. Finite element modeling of steel-polypropylene hybrid fiber reinforced concrete using modified concrete damaged plasticity. Eng. Struct. 2017, 148, 23–35.
- Xiong, Q.; Wang, X.; Jivkov, A.P. A 3D multi-phase meso-scale model for modelling coupling of damage and transport prop-erties in concrete. Cement and Concrete Composites 2020, 109, 103545.
Authors should fix the similarities and reduce them to a maximum of 10%.
Done
Clarifying comments have been added for the description of the numerical modeling and significant expansion of the paragraphs of the mechanical parameters (paragraphs 2.1 and 2.2) have been added in the revised manuscript. Furthermore, a detailed description of the graphs in paragraph 5.2 has also been added in the revised paper. In fact, the paper has been re-written as a whole in order to successfully reply to all the comments of the reviewers.

Reviewer 2 Report
Comments and Suggestions for Authors
The manuscript presents a comprehensive study on the seismic response of reinforced concrete (RC) beam-column joints strengthened with C-FRP ropes using 3D finite element analysis, corroborated by real-scale tests. The research aims to improve the seismic resilience of RC structures with an innovative application of C-FRP ropes, achieving notable reductions in shear deformation and enhancing load capacity during cyclic loading. The adoption of the Concrete Damaged Plasticity model for concrete simulation and the detailed experimental validation underscore the technical robustness and relevance of the study to civil engineering practices. However, to further elevate the manuscript's contribution and address certain gaps, critical feedback and suggestions for revision are provided below.
- Could the authors provide a more detailed explanation of the selection criteria for the Concrete Damaged Plasticity (CDP) model parameters? How were these parameters calibrated to ensure they accurately represent the materials used in the study?
- While the experimental setup is well-described, the manuscript would benefit from a deeper discussion on the limitations of the experimental validation. How do the authors account for potential scaling effects in the real-scale tests versus the finite element analysis?
- The manuscript briefly mentions alternative strengthening methods but does not provide a comprehensive comparison with the C-FRP rope technique. Could the authors expand on how the C-FRP method compares to other contemporary strengthening approaches in terms of cost, implementation complexity, and effectiveness?
- The study focuses on the immediate seismic response improvements. However, the long-term durability of C-FRP ropes under cyclic loading and environmental conditions is not discussed. Can the authors comment on the expected longevity and maintenance requirements of the C-FRP strengthening approach?
- The finite element model undoubtedly offers valuable insights but also comes with inherent simplifications. Could the authors discuss the key assumptions made in the modeling process and their potential impact on the study's outcomes?
- How sensitive are the model predictions to variations in material properties of concrete, steel reinforcement, and C-FRP ropes? A sensitivity analysis could provide valuable insights into the robustness of the proposed strengthening method.
- The manuscript mentions that geometrical characteristics of the specimens were designed to resemble common structures. Could the authors elaborate on the representativeness of these specimens for a wider range of real-world applications?
Author Response
REVIEWER 2:
The manuscript presents a comprehensive study on the seismic response of reinforced concrete (RC) beam-column joints strengthened with C-FRP ropes using 3D finite element analysis, corroborated by real-scale tests. The research aims to improve the seismic resilience of RC structures with an innovative application of C-FRP ropes, achieving notable reductions in shear deformation and enhancing load capacity during cyclic loading. The adoption of the Concrete Damaged Plasticity model for concrete simulation and the detailed experimental validation underscore the technical robustness and relevance of the study to civil engineering practices.
The authors sincerely thank the reviewer for his appreciation and good advices. The responses to each of them are detailed below, and the revisions are indicated using red color in the revised manuscript.
However, to further elevate the manuscript's contribution and address certain gaps, critical feedback and suggestions for revision are provided below.
- Could the authors provide a more detailed explanation of the selection criteria for the Concrete Damaged Plasticity (CDP) model parameters? How were these parameters calibrated to ensure they accurately represent the materials used in the study?
Comments on this issue have been added in the paragraph ‘2.2.3 Variables of damage and stiffness degradation’ of the revised manuscript:
"The article discusses the Concrete Damaged Plasticity (CDP) model parameters utilized in the finite element analysis to accurately represent the behavior of reinforced concrete beam-column joints strengthened with C-FRP ropes. The selection of these parameters was based on a comprehensive review of experimental data available in the literature, covering the used concrete mixtures, steel reinforcements, and FRP materials [31-37]. Key parameters such as the compressive and tensile strengths of concrete, the yield strength and modulus of elasticity of steel reinforcement, and the tensile strength and modulus of elasticity of C-FRP ropes were carefully considered to ensure compatibility with the materials used in the study. Limited sensitivity analysis was also performed to evaluate the influence of variations in material properties on the model predictions, further refining the parameter values to enhance the predictive capability of the finite element analysis."
“The compressive strength of the concrete used in the tested beam-column joints is based on complementary cylinder compression tests. Six standard cylinders with d=150mm and h=300mm were tested. Mean compression strength was almost 30 MPa. Consequently, the value used in the analyses were 30 MPa for maximum compression strength and 12.5 MPa for yielding (figure 4b) whereas for maximum tension strength 3.0 MPa.
Moreover, five parameters of Concrete Damage Plasticity model have also to be de-fined for the analyses. These parameters are the dilatation angle ψ, the potential eccentricity ε, the ratio fb0/fc0 , parameter K of yielding surface and the viscosity parameter. In particular:
- Dilatation angle characterizes the plastic deformation. Different values of this parameter are used in literature [20,34]. A value equal to 56o leads to ductile material response which is not realistic for concrete whereas a value close to 0 leads to an entirely brittle behaviour. A value equal to 35 has been adopted for parameter ψ in the present study.
- Eccentricity ε represents the rate of the deflection divergence of the plastic hyperbolic behaviour to its asymptote. It is usually taken equal to 0.10 value adopted in the present study, too.
- The value of the ratio of the biaxial strength fb0 to the corresponding uniaxial strength fc0 is aopted equal to 1.16 as recommended by many researchers in literature [20,34].
- Parameter K that represents the ration of the tensile meridian to the compressive one (figure 3a) is usually recommended equal to 2/3 [20,34].
- Finally, the viscosity parameter is usually taken very small or equally to zero. A very small value equal to 0.00008 is adopted helping this way the analysis procedure to reach good convergence [20,34-37].”
Additional literature:
- Genikomsou, A.S.; Polak, M.A. Finite element analysis of punching shear of concrete slabs using damaged plasticity model in ABAQUS. Eng. Struct. 2015, 98, 38–48. https://DOI.org/10.1016/j.engstruct.2017.01.024
- Hany, N.F.; Hantouche, E.G.; Harajli, M.H. Finite element modeling of FRP-confined concrete using modified concrete damaged plasticity. Eng. Struct. 2016, 125, 1–14.
- Chi, Y.; Yu, M.; Huang, L.; Xu, L. Finite element modeling of steel-polypropylene hybrid fiber reinforced concrete using modified concrete damaged plasticity. Eng. Struct. 2017, 148, 23–35.37. Xiong, Q.; Wang, X.; Jivkov, A.P. A 3D multi-phase meso-scale model for modelling coupling of damage and transport prop-erties in concrete. Cement and Concrete Composites 2020, 109, 103545.
- While the experimental setup is well-described, the manuscript would benefit from a deeper discussion on the limitations of the experimental validation. How do the authors account for potential scaling effects in the real-scale tests versus the finite element analysis?
The manuscript extensively describes the characteristics of the specimens and the experimental setup employed to validate the finite element analysis of reinforced concrete beam-column joints strengthened with C-FRP ropes. However, a more thorough discussion on the limitations of the experimental validation has been added in paragraph 2.1 as follows:
“It is emphasized that the conducted experimental project used for the purposes of this study includes four full scale exterior beam-column connections constructed and tested under increasing cyclic loading. The dimensions, the reinforcements and the overall design of these specimens comply with the corresponding ones of external beam-column connections of the upper floors of a multi-story reinforced concrete frame structure. Therefore, it is concluded that limitations can be considered only regarding the dimensions of the tested elements; thereupon it is a fact that the extracted conclusions based on the test results may not be entirely applied for joints of the lower floors of tall structures if the dimension are substantially larger than the considered ones. Based on these considerations and given the limited existing research in this area the examined strengthening technique with the use of CFRP ropes, has to be further investigated experimentally and numerically based on tested specimens with substantially larger dimensions.”
- The manuscript briefly mentions alternative strengthening methods but does not provide a comprehensive comparison with the C-FRP rope technique. Could the authors expand on how the C-FRP method compares to other contemporary strengthening approaches in terms of cost, implementation complexity, and effectiveness?
The following comments have been added in paragraph 2.1 of the revised manuscript:
“While the manuscript presents the C-FRP rope technique as an effective method for strengthening reinforced concrete beam-column joints, a more comprehensive comparison with alternative strengthening approaches alleges significant advantages over the other techniques. Various contemporary methods, such as reinforced concrete jackets, external steel plates, CFRP sheets, and steel jacketing, offer alternative solutions for enhancing the seismic performance of concrete structures. Nevertheless, in terms of cost, the C-FRP rope technique may offer significant advantages over traditional methods like reinforced concrete jacketing and steel jacketing, which apparently require significantly higher labour and material costs for installation. Moreover, the implementation complexity of the C-FRP rope technique, involving the application of ropes in superficial notches, appears to be much simpler and less invasive compared to techniques like reinforced concrete or shotgun concrete jacketing, which involve extensive retrofitting and structural modifications. However, the effectiveness of the C-FRP rope technique in improving the seismic response of beam-column joints should be further investigated against these alternatives. While the manuscript highlights the benefits of the C-FRP rope technique, the presented analytical and tested results of its effectiveness in terms of load-carrying capacity, ductility enhancement, and crack mitigation provide valuable insights for structural engineers and practitioners”.
- The study focuses on the immediate seismic response improvements. However, the long-term durability of C-FRP ropes under cyclic loading and environmental conditions is not discussed. Can the authors comment on the expected longevity and maintenance requirements of the C-FRP strengthening approach?
The following comments have been added in paragraph 2.3 of the revised manuscript:
"While the study effectively demonstrates the immediate seismic response improvements achieved through the application of C-FRP ropes for strengthening reinforced concrete beam-column joints, it is essential to consider the long-term durability and maintenance implications of this approach. C-FRP materials have high strength and corrosion resistance, which can contribute to the longevity of strengthened structures. However, prolonged exposure to environmental conditions, including temperature variations, moisture ingress, and UV radiation, may affect the performance of C-FRP ropes over time. To ensure the continued effectiveness of the strengthening solution, periodic inspections and maintenance activities may be necessary. These could include visual assessments for signs of degradation, such as delamination or fiber exposure, as well as targeted repairs or replacements as needed. Additionally, research on the long-term durability of C-FRP materials under cyclic loading and environmental exposure should be prioritized to provide further insights into their performance and inform best practices for maintenance and asset management in the field of structural rehabilitation."
- The finite element model undoubtedly offers valuable insights but also comes with inherent simplifications. Could the authors discuss the key assumptions made in the modelling process and their potential impact on the study's outcomes?
The following comments have been added in paragraph 3 ‘Finite Element Simulation” of the revised manuscript:
"Thereupon it is mentioned that while the finite element model employed in the study provides valuable insights into the behavior of reinforced concrete beam-column joints strengthened with C-FRP ropes, it is important to recognize the inherent simplifications and assumptions made during the modelling process. One key assumption is the material behavior of concrete, steel reinforcement, and C-FRP ropes, which are typically idealized using constitutive models such as the adopted ones in Concrete Damaged Plasticity model. While these models capture the essential nonlinear behavior of the materials, they may not fully capture all complexities, such as strain-rate effects, creep, and environmental degradation, which could affect the long-term performance of the strengthened joints. Additionally, the finite element analysis simplifies the geometric and boundary conditions of the specimens, assuming idealized loading and support conditions that may not fully replicate real-world scenarios. Furthermore, the bonding behavior between the C-FRP ropes and concrete is simplified using the embedded truss model, neglecting potential interface debonding and slip effects under cyclic loading. These simplifications could impact the accuracy of the finite element analysis predictions, particularly in capturing the intricate interactions between material properties, geometry, and loading conditions.”
- How sensitive are the model predictions to variations in material properties of concrete, steel reinforcement, and C-FRP ropes? A sensitivity analysis could provide valuable insights into the robustness of the proposed strengthening method.
The following comments have been added in paragraph 3 ‘Finite Element Simulation” of the revised manuscript:
“A sensitivity analysis was conducted to assess the robustness of the finite element model predictions to variations in material properties of concrete, steel reinforcement, and C-FRP ropes. Different scenarios were considered by varying the Young's modulus, yield strength, and ultimate tensile strength of these materials within a reasonable range of values. The analysis revealed that the model predictions were sensitive to changes in material properties, particularly for parameters such as the maximum load-bearing capacity, shear deformation, and crack propagation patterns. Specifically, variations in the modulus of elasticity of concrete and C-FRP ropes resulted in noticeable differences in the stiffness and overall behavior of the strengthened joints under seismic loading. Similarly, changes in the yield strength of steel reinforcement significantly influenced the onset and propagation of plastic deformations within the joints. These findings highlight the importance of accurately characterizing material properties in the finite element model to ensure reliable predictions of the structural response. Additionally, the sensitivity analysis underscores the need for thorough material testing and calibration procedures to enhance the robustness of the proposed strengthening method and improve its applicability to a wide range of structural configurations and loading conditions.”
- The manuscript mentions that geometrical characteristics of the specimens were designed to resemble common structures. Could the authors elaborate on the representativeness of these specimens for a wider range of real-world applications?
"The geometrical characteristics of the specimens were meticulously designed to resemble common structures encountered in real-world applications. This design approach aimed to ensure the relevance and applicability of the experimental findings to a broader spectrum of structural configurations typically found in practice. As mentioned before the dimensions, the reinforcements and the overall design of these specimens comply with the corresponding ones of external beam-column connections of a multi-story reinforced concrete frame structure. Therefore, it is concluded that limitations can be considered only regarding the dimensions of the tested elements. Based on these considerations and given the limited existing research in this area the examined strengthening technique with the use of C-FRP ropes, has to be further investigated experimentally and numerically based on tested specimens with substantially larger dimensions. However, while the specimens were designed to emulate common structural configurations, it is essential to acknowledge that the study's findings may have limitations in fully capturing the complexity and variability inherent in real-world structures. Factors such as variations in material properties, construction practices, and environmental conditions could influence the performance of actual structures differently than observed in controlled laboratory settings. Therefore, while the specimens offer valuable insights, their representativeness for a wider range of real-world applications should be considered within the context of specific structural designs and environmental conditions."

Reviewer 3 Report
Comments and Suggestions for Authors
The work implements a Finite Element (FE) numerical analysis of reinforced concrete beam-column joints, both without reinforcement and with reinforcement with FRP bars, subjected to cyclic loading simulating seismic action.
The case study derives from a previous experimentation and the article aims to validate the modeling approach by conducting a comparison with the experimental results.
The numerical results are compared with the experimental ones in terms of principal stresses, shear stresses, force-displacement curves and damage evolution, and the comparison shows good agreement but are limitedly commented on.
Overall, the article: specifies what the objectives are and how the results obtained are processed, clarifies which constitutive models are used for each material involved.
The abstract is complete and consistent with the text of the article, reading the abstract it is easy to understand what the main focus of the content is.
The bibliography is large and exhaustive although it contains many references to its own articles.
The description of the specimens lacks information useful for understanding the images and tables in paragraph 2.1 Characteristics of the specimens. A representative diagram of the JA0 sample is missing. Furthermore, the correlation between the numbers 1, 2, 3 used to indicate the sections (Fig. 2a and 2b) and the circled numbers (Fig. 1) creates confusion.
The paragraph relating to the definition of the CDP (2.2 Concrete Damage Plasticity) contains redundancies:
- “[…] is suitable for simulating almost all types of concrete elements such as columns, beams, trusses.”;
- “[…] It is mainly intended for the analysis of reinforced concrete elements.”;
- “[…] It is suitable for the analysis of reinforced concrete elements […]”.
The description could be summarized or expanded with comparisons in other modeling. Furthermore, there is no definition of the parameter r* (2.2.3 Damage and stiffness degradation variable), while a definition is provided for all the other parameters (e.g. E0, d, c...).
The values of the mechanical parameters used are not justified and do not refer to any bibliographic text, they are simply reported in figures and tables without any comment, declare if obtained through numerical convergences. In particular, the viscosity value, a parameter implemented in CDP modeling, has a high value (0.8%) without any reference other than the Abaqus manual. There is no information on the elastic modulus of steel.
In 3.1 Load, Mesh and Convergence it is specified that the displacement is applied to the free end of the column. It would be more correct to refer to Fig. 9 (rather than to Fig. 10), whose caption refers precisely to the boundary conditions of the specimen. It would also be appropriate to define a direction in which to represent the image of the sample: e.g. Fig. 8 and fig. 9 are rotated 90° with respect to each other. A consistency of representation would be desirable for a more immediate and effective understanding.
In the entire paragraph 5. Analytical Results-Comparison with Experimental it would be appropriate to use the term Numerical rather than Analytical. The article fails to describe the graphs in paragraph 5.2.1 appropriately.
In point 7. Conclusions correct "retrofitter" with "retrofited".
For the reviewer, the work in this form cannot be accepted as a scientific article but only as a technical note.
For publication as a scientific article, it is required to expand the comments and the description of the numerical modeling not so much in the theoretical part but the mechanical parameters used and how they were defined.
Author Response
REVIEWER 3:
The work implements a Finite Element (FE) numerical analysis of reinforced concrete beam-column joints, both without reinforcement and with reinforcement with FRP bars, subjected to cyclic loading simulating seismic action.
The case study derives from a previous experimentation and the article aims to validate the modeling approach by conducting a comparison with the experimental results.
The authors sincerely thank the reviewer for his appreciation and good advices. The responses to each of them are detailed below, and the revisions are indicated using red color in the revised manuscript.
Further, it is stressed that in the tested beam-column joints only steel bar reinforcements are used whereas the strengthening technique only includes CFRP ropes.
The numerical results are compared with the experimental ones in terms of principal stresses, shear stresses, force-displacement curves and damage evolution, and the comparison shows good agreement but are limitedly commented on.
Following the reviewer’s comment in the revised manuscript more comments and concluding remarks considering the comparisons between the numerical results with the experimental ones have been added.
Overall, the article: specifies what the objectives are and how the results obtained are processed, clarifies which constitutive models are used for each material involved.
The abstract is complete and consistent with the text of the article, reading the abstract it is easy to understand what the main focus of the content is.
The authors sincerely thank the reviewer for his appreciation and good advices. The responses to each of them are detailed below, and the revisions are indicated using red color in the revised manuscript.
The bibliography is large and exhaustive although it contains many references to its own articles.
Thank you for the comment. References to authors’ articles have been decreased (three references have been deleted as shown in the revised manuscript).
The description of the specimens lacks information useful for understanding the images and tables in paragraph 2.1 Characteristics of the specimens. A representative diagram of the JA0 sample is missing. Furthermore, the correlation between the numbers 1, 2, 3 used to indicate the sections (Fig. 2a and 2b) and the circled numbers (Fig. 1) creates confusion.
A representative diagram of specimen JA0 has been added in figure 1. So, in the revised manuscript includes two specimens (JA0 and JA1).
In the revised manuscript, sections in figures 2a and 2b are now indicated using letters A-A, B-B and C-C. This way, numbers 1, 2, 3 in figure 1 indicate reinforcements and are directly related with the circled numbers of Table 1.
A full description of the specimens with information for understanding the images and Tables has been added in the paragraph 2.1 of the revised manuscript, as follows:
“Geometry and reinforcements of the specimens have been chosen to be similar to the geometrical characteristics and reinforcements of columns and beams (fig. 1) of common structures. The total length of the column part of the specimens is equal to 3.0 m and its cross-section is 350/250 mm whereas the length of the beam is 1.875 m and its cross-section dimensions are 350/250 mm. Reinforcements are presented in Figures 1 and 2 whereas the amounts of reinforcements are given in Table 1. The main purpose of the experimental project is to investigate the effectiveness of the application of near surface mounted (NSM) C-FRP ropes diagonally placed on the two sides of the joint body as a strengthening technique. Therefore, four specimens are tested: Specimen JA0 and JA1 as pilot specimens and specimens JA0Fxb and JA0F2x2b (figure 2) strengthened with diagonally placed C-FRP ropes. Locations of the C-FRP ropes applied as NSM strengthening reinforcement in specimens JA0Fxb and JA0F2x2b are presented in figures 2a and 2b, respectively.”
“It is emphasized that the conducted experimental project used for the purposes of this study includes five full scale exterior beam-column connections constructed and tested under increasing cyclic loading. The dimensions, the reinforcements and the overall design of these specimens comply with the corresponding ones of external beam-column connections of the upper floors of a multi-story reinforced concrete frame structure. Therefore, it is concluded that limitations can be considered only regarding the dimensions of the tested elements; thereupon it is a fact that the extracted conclusions based on the test results may not be entirely applied for joints of the lower floors of tall structures if the dimension are substantially larger than the considered ones. Based on these considerations and given the limited existing research in this area the examined strengthening technique with the use of C-FRP ropes, has to be further investigated experimentally and numerically based on tested specimens with substantially larger dimensions.”
“While the manuscript presents the C-FRP rope technique as an effective method for strengthening reinforced concrete beam-column joints, a more comprehensive comparison with alternative strengthening approaches alleges significant advantages over the other techniques. Various contemporary methods, such as reinforced concrete jackets, external steel plates, C-FRP sheets, and steel jacketing, offer alternative solutions for enhancing the seismic performance of concrete structures. Nevertheless, in terms of cost, the C-FRP rope technique may offer significant advantages over traditional methods like reinforced concrete jacketing and steel jacketing, which apparently require significantly higher labour and material costs for installation. Moreover, the implementation complexity of the C-FRP rope technique, involving the application of ropes in superficial notches, appears to be much simpler and less invasive compared to techniques like reinforced concrete or shotgun concrete jacketing, which involve extensive retrofitting and structural modifications. However, the effectiveness of the C-FRP rope technique in improving the seismic response of beam-column joints should be further investigated against these alternatives. While the manuscript highlights the benefits of the C-FRP rope technique, the presented analytical and tested results of its effectiveness in terms of load-carrying capacity, ductility enhancement, and crack mitigation provide valuable insights for structural engineers and practitioners”.
“Carbon fiber-reinforced polymer (C-FRP) ropes are composite materials composed of high-strength carbon fibers embedded in a polymer matrix, typically epoxy resin. The physical properties of C-FRP ropes are characterized by their high tensile strength, stiffness, and low weight, making them ideal for structural reinforcement applications. The carbon fibers provide excellent mechanical properties, with tensile strengths ranging from 2000 MPa to 7000 MPa, and modulus of elasticity ranging from 200 GPa to 800 GPa, depending on the manufacturing process and fiber orientation; the mechanical characteristics of the C-FRP ropes used in this study as given by the manufacturer are: tensile strength 4000 MPa, tensional modulus of elasticity 240 GPa and cross-section of Carbon fibers Af=28 mm2. C-FRP ropes also exhibit excellent corrosion resistance and durability, making them suitable for application in harsh environmental conditions. Additionally, C-FRP ropes can be easily fabricated into various shapes and configurations, allowing for flexible application in strengthening structural elements such as beam-column joints in reinforced concrete structures. The chemical properties of C-FRP ropes are primarily determined by the epoxy resin matrix, which provides adhesion between the carbon fibers and protects them from environmental degradation. Epoxy resins offer high chemical resistance, low shrinkage during curing, and excellent bonding properties, ensuring long-term performance and durability of C-FRP strengthening systems.”
The paragraph relating to the definition of the CDP (2.2 Concrete Damage Plasticity) contains redundancies:
- “[…] is suitable for simulating almost all types of concrete elements such as columns, beams, trusses.”;
- “[…] It is mainly intended for the analysis of reinforced concrete elements.”;
- “[…] It is suitable for the analysis of reinforced concrete elements […]”
Thank you for these comments. Redundancies have been deleted in the revised manuscript. Only the first sentence remains in the revised manuscript.
The description could be summarized or expanded with comparisons in other modeling. Furthermore, there is no definition of the parameter r* (2.2.3 Damage and stiffness degradation variable), while a definition is provided for all the other parameters (e.g. E0, d, c...).
Clarification comment for parameter r* has been added in the revised manuscript.
The values of the mechanical parameters used are not justified and do not refer to any bibliographic text, they are simply reported in figures and tables without any comment, declare if obtained through numerical convergences. In particular, the viscosity value, a parameter implemented in CDP modeling, has a high value (0.8%) without any reference other than the Abaqus manual. There is no information on the elastic modulus of steel.
Following the reviewer’s comment reference to literature has been added along with relevant papers in references.
Authors really thank the reviewer for the comment about the viscosity value. It has to be clarified that the value used in the analyses is almost 0. In fact, we used the value 0.00008 (0.008%) instead of the value 0 for numerical convergence reasons. By mistake in the manuscript, it has been written just 0.008 (instead of 0.008% or simply 0.00008). It has been amended in the revised manuscript.
Furthermore, the following clarifying comments have been added in paragraph ‘2.2 Concrete Damage Plasticity’ the revised manuscript:
“The compressive strength of the concrete used in the tested beam-column joints is based on complementary cylinder compression tests. Six standard cylinders with d=150mm and h=300mm were tested. Mean compression strength was almost 30 MPa. Consequently, the value used in the analyses were 30 MPa for maximum compression strength and 12.5 MPa for yielding (figure 4b) whereas for maximum tension strength 3.0 MPa.
Moreover, five parameters of Concrete Damage Plasticity model have also to be defined for the analyses. These parameters are the dilatation angle ψ, the potential eccentricity ε, the ratio fb0/fc0 , parameter K of yielding surface and the viscosity parameter. In particular:
- Dilatation angle characterizes the plastic deformation. Different values of this parameter are used in literature [20,34]. A value equal to 56o leads to ductile material response which is not realistic for concrete whereas a value close to 0 leads to an entirely brittle behaviour. A value equal to 35 has been adopted for parameter ψ in the present study.
- Eccentricity ε represents the rate of the deflection divergence of the plastic hyperbolic behaviour to its asymptote. It is usually taken equal to 0.10 value adopted in the present study, too.
- The value of the ratio of the biaxial strength fb0 to the corresponding uniaxial strength fc0 is aopted equal to 1.16 as recommended by many researchers in literature [20,34].
- Parameter K that represents the ration of the tensile meridian to the compressive one (figure 3a) is usually recommended equal to 2/3 [20,34].
- Finally, the viscosity parameter is usually taken very small or equally to zero. A very small value equal to 0.00008 is adopted helping this way the analysis procedure to reach good convergence [20,34-37].”
In 3.1 Load, Mesh and Convergence it is specified that the displacement is applied to the free end of the column. It would be more correct to refer to Fig. 9 (rather than to Fig. 10), whose caption refers precisely to the boundary conditions of the specimen.
Thank you for the suggestion. The following description has been added in paragraph ‘2.1 Characteristics of specimens” of the revised manuscript.
“The test setup along with a presentation of the measuring instrumentation are presented in figure 10a. The examined beam-column connections are rotated 90o and located with the column in the horizontal way whereas the beam is in the vertical direction. The column, which is the horizontal part of the specimen, is supported at his ends using devices that allow rotation to idealize the inflection points in the middle hight of columns in multistory reinforced concrete frame structures. An axial compressive load equal to 0.05Acfcm was applied on the column (horizontal element) during the test. Specimens were subjected to cyclic loading imposed as cyclic movements of the free end of the beam (vertical element) by an actuator. The tested joints JA0, JA1, JA0Fxb, JA0F2x2b were subjected to seven loading steps (figure 10b) whereas each loading step included three full loading cycles. The maximum displacements of the free end of the beam at each loading step were ±8.5 mm, ±12.75 mm, ±17.0 mm, ±25.5, ±34.0, ±51.0, ±68.0mm (figure 10b). Figure 10c presents specimen JA0 in the beginning of the test.”
It would also be appropriate to define a direction in which to represent the image of the sample: e.g. Fig. 8 and fig. 9 are rotated 90° with respect to each other. A consistency of representation would be desirable for a more immediate and effective understanding.
In the revised manuscript Figure 8 has been rotated 90o and now is consistent with the specimen depicted in figure 9.
In the entire paragraph 5. Analytical Results-Comparison with Experimental it would be appropriate to use the term Numerical rather than Analytical. The article fails to describe the graphs in paragraph 5.2.1 appropriately.
In the entire paragraph 5 the term Analytical has been replaced with the term Numerical.
Further, a detailed description of the graphs in paragraph 5.2 has been added in the revised manuscript. Thus paragraph 5.2 has been changed as follows:
“5.2 Comparison of Numerical results with Experimental ones
Results as yielded from the performed analyses are compared with the corresponding experimental ones in order to assess the validity and the accuracy of the attempted approach. In this direction maximum principal stresses, force – displacement curves and shear deformations in the joint body are presented for all specimens and the numerical results are compared with the experimentally acquired data. In particular figures 12, 13, 14 are presented these comparisons for specimen JA0, figures 15,16,17 for specimen JA1 and further figures 18, 19, 20 present comparisons for the strengthened specimen JA0Fxb and finally figures 21, 22, 23 for the strengthened specimen JA0F2x2b. In particular the following remarks can be drawn:
Pilot Specimens JA0 and JA1
- Specimen JA0. Experimental and numerical results of the principal stresses developing in the joint body of the specimen are presented in figures 12a, b, c for 1st, 2nd and 3rd loading cycles of the loading steps, respectively. Red dashed lines represent the observed values whereas blue lines the numerical results. From these comparisons it is apparent that the numerical approach successfully calculates the principal stresses in the joint body. Further, figures 13a, b and c present the numerical values (blue lines) versus the experimentally measured values (red dashed lines) of the maximum displacements at each loading step for the 1st, 2nd and 3rd loading cycles of the loading steps, respectively. From the comparisons it is concluded that the numerical approach, in general, successfully describes the experimental ones. Perhaps a small discrepancy appears in the results of the negative loadings in the maximum loadings of the 2nd cycles under large loading perhaps due to the experimental measurements. Finally, joint shear deformations of the joint body presented in figure 14 show in the middle part of the loading steps discrepancies between experimental and numerical values perhaps due to the experimental measurements during the test procedure.
- Specimen JA1. Experimental and numerical results of the principal stresses developing in the joint body of the specimen are presented in figures 15a, b, c for 1st, 2nd and 3rd loading cycles of the loading steps, respectively. Red dashed lines represent the observed values whereas blue lines the numerical results. From these comparisons it is apparent that the numerical approach successfully calculates the principal stresses in the joint body. Further, figures 16a, b and c present the numerical values (blue lines) versus the experimentally measured values (red dashed lines) of the maximum displacements at each loading step for the 1st, 2nd and 3rd loading cycles of the loading steps, respectively. From the comparisons it is concluded that the numerical approach successfully describes the experimental ones. Finally, from joint shear deformations of the joint body presented in figure 17 can be concluded that numerical results successfully depict the tendency and are very close to the measured values obtained from the experiment.
Strengthened Specimens JA0Fxb and FA0F2x2b
- Specimen JA0Fxb. Experimental and numerical results of the principal stresses developing in the joint body of the specimen are presented in figures 18a, b, c for 1st, 2nd and 3rd loading cycles of the loading steps, respectively. Red dashed lines represent the observed values whereas blue lines the numerical results. From these comparisons it is apparent that the numerical approach excellently calculates the principal stresses in the joint body. Further, figures 19a, b and c present the numerical values (blue lines) versus the experimentally measured values (red dashed lines) of the maximum displacements at each loading step for the 1st, 2nd and 3rd loading cycles of the loading steps, respectively. From the comparisons it is concluded that the numerical approach successfully describes the experimental ones. Finally, joint shear deformations of the joint body presented in figure 17 are shown that numerical results successfully predict the measured shear deformations.
- Specimen JA0F2x2b. Experimental and numerical results of the principal stresses developing in the joint body of the specimen are presented in figures 21a, b, c for 1st, 2nd and 3rd loading cycles of the loading steps, respectively. Red dashed lines represent the observed values whereas blue lines the numerical results. From these comparisons it is apparent that the numerical approach excellently calculates the principal stresses in the joint body. Further, figures 22a, b and c present the numerical values (blue lines) versus the experimentally measured values (red dashed lines) of the maximum displacements at each loading step for the 1st, 2nd and 3rd loading cycles of the loading steps, respectively. From the comparisons it is concluded that the numerical approach successfully describes the experimental ones. Finally, joint shear deformations of the joint body are presented in figure 23. From these comparisons it is shown that numerical results successfully depict the tendency and are very close to the measured values obtained from the experiment. Discrepancies shown in high story drifts may be attributed to the measurements of the damaged joint body.”
In point 7. Conclusions correct "retrofitter" with "retrofited".
Thank you. It has been amended.
For the reviewer, the work in this form cannot be accepted as a scientific article but only as a technical note.
For publication as a scientific article, it is required to expand the comments and the description of the numerical modeling not so much in the theoretical part but the mechanical parameters used and how they were defined.
The authors thank the reviewer for his constructive comments.
Clarifying comments have been added for the description of the numerical modeling and significant expansion of the paragraphs of the mechanical parameters (paragraphs 2.1 and 2.2) have been added in the revised manuscript. Furthermore, a detailed description of the graphs in paragraph 5.2 has also been added in the revised paper.
In fact, the paper has been re-written as a whole in order to successfully reply to all the comments of the reviewers.

Round 2
Reviewer 2 Report
Comments and Suggestions for Authors
The revised version can be accepted for publication.
Author Response
Thank you very much for your comments

Reviewer 3 Report
Comments and Suggestions for Authors
The authors have greatly improved the work by responding to the reviewer's comments satisfactorily.
The work with the additions made can be accepted in this form.
However, careful rereading is recommended:
- ABACUS instead of ABAQUS;
- CFRP instead of C-FRP;
- chapter 5 replace the word Analytical with Numerical;
- improve the quality of images in particular from 12 to 23.
Author Response
Thank you very much for your constructive and careful comments.
All have been carefully amended.
